# Viral-bacterial co-infections screen in vitro reveals molecular processes affecting pathogen proliferation and host cell viability

Philipp Walch ⓘ & Petr Broz ⓘ ✉

The broadening of accessible methodologies has enabled mechanistic insights into single-pathogen infections, yet the molecular mechanisms underlying co-infections remain largely elusive, despite their clinical frequency and relevance, generally exacerbating symptom severity and fatality. Here, we describe an unbiased in vitro screening of pairwise co-infections in a murine macrophage model, quantifying pathogen proliferation and host cell death in parallel over time. The screen revealed that the majority of interactions are antagonistic for both metrics, highlighting general patterns depending on the pathogen virulence strategy. We subsequently decipher two distinct molecular interaction points: Firstly, murine Adenovirus 3 modifies ASC-dependent inflammasome responses in murine macrophages, altering host cell death and cytokine production, thereby impacting secondary *Salmonella* infection. Secondly, murine Adenovirus 2 infection triggers upregulation of Mprip, a crucial mediator of phagocytosis, which in turn causes increased *Yersinia* uptake, specifically in virus pre-infected bone-marrow-derived macrophages. This work therefore encompasses both a first-of-its-kind systematic assessment of host-pathogen-pathogen interactions, and mechanistic insight into molecular mediators during co-infection.

A major risk factor in the clinical outcome of infectious diseases is the occurrence of secondary infections with another pathogen. This has been shown most predominantly for patients suffering from infection with Human Immunodeficiency Virus (HIV). Patients have a massively increased susceptibility to super-infections with a second pathogen, which can be viral[1], bacterial[2], fungal[3] or of another nature[4]. Furthermore, other infectious diseases display increased clinical severity due to the occurrence of super-infections, such as Influenza A virus (IAV), where, staphylococcal or streptococcal co-infections increase disease severity and fatality and are commonly isolated from hospitalized patients[5]. Besides that, Rotavirus and Norovirus have been described as clinically relevant infectious agents that occur with co-infections of other enteric pathogens, such as *Escherichia coli*, which triggers exacerbated symptoms in children[6,7], or *Clostridium difficile*, causing an increased bacterial burden[8,9].

Conceptually, co-infecting pathogens can impact each other on various levels, highlighting the complexity of the perturbations that they introduce to the host[10]. While HIV exerts its role in co-infections on a systemic level, i.e. through the depletion of an essential compartment of the patient's immune response, cellular interaction points, such as pathogen adherence or invasion, immune detection, cell death induction and signaling to neighboring cells have also been hypothesized[11].

The understanding of host-pathogen interactions in the case of single-pathogen infections on a cellular and molecular level continues to broaden. This is due to the elucidation of molecular targets for pathogenic effectors[12–16], the identification of host signaling cascades within the innate immune response[17–20], as well as the characterization of limited resources triggering competition between the pathogen and the host during infection[21,22]. These studies have played a vital part in developing new strategies to intercept infection routes, disrupt

University of Lausanne, Department of Immunobiology, Chemin des Boveresses 155, CH-1066 Epalinges, Switzerland. ✉e-mail: petr.broz@unil.ch

pathogen spread and thereby combat infectious diseases at various levels. Yet the same level of mechanistic insight is lacking for co-infections of two different pathogens, the study of which has so far mainly occurred on the organismal or systemic level[11].

Some studies have pioneered the molecular understanding of the interplay between different pathogens, such as IAV and *Streptococcus pneumoniae*[5,23] or HIV and *Mycobacterium tuberculosis*[24,25]. Those studies show a multitude of pathogen-pathogen dependencies, ranging from immune priming[23,26] to resource limitation within the host cell[27–29]. Still, a comprehensive picture of the interplay between the co-infecting pathogens, as well as a detailed understanding of the role of host innate immune response remains elusive. Furthermore, while the examples mentioned above lead to exacerbation of the infection and increased bacterial proliferation, there have also been several reports of antagonistic interactions between two pathogens, such as coinfections of *Wolbachia* and Dengue virus[30,31] or other Flaviviridae[27]. Due to the clinical impact of co-infections and the current lack of in-depth knowledge, it is essential that we apply the same molecular and immunological techniques that are used to probe the host-pathogen interface, to elucidate the co-dependency and interactions between co-occurring pathogens.

To address this sparsity of knowledge, we assessed infection with murine Adenovirus (mAdV) or murine Norovirus (MNV) in pairwise combination with a panel of enteropathogenic bacteria, spanning *Salmonella*, *Shigella*, *Citrobacter*, *Vibrio*, *Yersinia* and *Escherichia*. Rather than focusing solely on the pathogens, we further investigated the role of the host cell, especially with respect to the induction of cell death. To do so, we established an in vitro screening setup to monitor co-infections in high-throughput and in parallel, by scoring pathogen growth and host cell death using an automated plate reader-based quantification. We chose the in vitro setup in order to disentangle interactions occurring at the cellular level, and to be able to specifically perturb the experimental model. Furthermore, an in vitro study allows for the unbiased testing of various combinations in parallel. We considered several possible host cell models for the screening approach, and selected RAW264.7 macrophages for the following reasons: 1) RAW264.7 are a well-established cell culture model in the field of infection biology[32], 2) their growth properties, adherence and maintainability in culture allow for the throughput needed in this study, and reduce the technical noise in the plate reader, 3) they are permissive for all the viruses in the study, and can engulf bacterial pathogens, or be actively invaded and used as replicative niche[12].

On the pathogen side, we selected murine viral pathogens that were expandable in cell culture, and which represent different strains, ranging from those causing acute infection (such as mAdV1, mAdV2 and MNV1) to chronic colonization (mAdV3, MNV2, MNV3 and MNV$_{CR3}$)[33–35]. The bacterial pathogens were chosen to represent different lifestyles during infection: *Salmonella* and *Shigella* possessing the ability to actively invade macrophages and to use them as proliferative niche[36], *Yersinia* specifically translocating effectors to resist uptake by the host cell[37], and *Citrobacter*, *Escherichia* and *Vibrio* adhering to the host cell to inject toxic effector proteins through their secretion systems[38]. Besides, all bacterial pathogens are Gram-negative, allowing for a control treatment with the surface molecule lipopolysaccharide (LPS).

In this work, we present an unbiased overview over synergistic and antagonistic interactions with respect to host cell death and pathogenic proliferative behavior or clearance, using a Bliss score model. We show the reliability of the data by successful validation using orthogonal biochemical assays and highlight two vignettes of new biology, which were uncovered by the screen. Firstly, we demonstrate how mAdV3, in addition to targeting previously implicated cellular processes, such as endocytosis and the regulation of the cell cycle, directly influences the ability of the host to respond to inflammasome stimuli at the level of Apoptosis-associated speck-like protein containing a CARD (ASC). We could link this alteration of the host cell's innate immune response to secondary phenotypic outcomes, such as reduced cell death and cytokine production, which in turn has a major impact on secondary bacterial infections, especially for pathogens, which utilize host macrophages as replicative niche. Secondly, we characterize the impact that mAdV2 plays during secondary infection with *Yersinia*. We present a large dataset of proteins altered upon Adenovirus infection and show that upregulation of Myosin phosphatase Rho-interacting protein (Mprip) in response to the virus results in an increased uptake of the bacterial pathogen, specifically in virally infected cells.

This study thereby presents the community with a broad dataset of pairwise interactions during viral-bacterial co-infection, and furthermore expands our mechanistic understanding of the host-pathogen-pathogen interface.

## Results

### A parallelized, unbiased screen reveals diverse viral-bacterial interactions during host co-infection

To disentangle the interaction between two co-infecting pathogens in infected cells, we assembled a panel of seven viral (murine Adenovirus (mAdV) 1, 2 and 3, and murine Norovirus (MNV) 1, 2, 3 and CR3) and six bacterial (*Salmonella enterica* serovar Typhimurium, *Shigella flexneri*, *Yersinia enterocolitica*, *Citrobacter rodentium*, *Vibrio cholerae* and pathogenic *Escherichia coli*) enteric pathogens (Fig. 1A, I). As an experimental system, we chose murine macrophages (RAW264.7), since macrophages play a major role in the restriction or propagation of systemic disease, and have been reported to either interact with or respond to all 13 pathogens in our panel[39–48]. Furthermore, RAW264.7 are a well-described and recognized host-cell model and can be grown in sufficient numbers to allow for the throughput needed in the screen.

We first confirmed that RAW264.7 cells were indeed infected by each of the pathogens by performing infections with single fluorescently labeled pathogens, assessing pathogen load by measuring fluorescence intensity, and host cell viability by measuring Propidium Iodide (PI) uptake over time (Figure S1). This analysis showed variable levels of proliferation and host cell death induction for the different pathogens, with bacteria causing in general more robust induction of host cell death. Moreover, the analysis of single infections allowed us to define appropriate Multiplicities of infection (MOIs) and timepoints for the co-infection study, i.e. avoiding excessive viral-induced cell death in the 'subsequent infection' setting (see below).

Having confirmed that each viral and bacterial pathogen in our panel was able to infect macrophages, we next assessed each pairwise combination in a parallelized and unbiased manner by measuring pathogen proliferation or clearance (Fig. 1A, II), as well as host cell death, determined by PI uptake over time, and comparing these to the single pathogen infections (Figure S1 or deposited datasets on Mendeley Data https://data.mendeley.com/datasets/thjzhzdpvc/1, as well as Source Data File). Since infections can be highly dynamic, we compared 'subsequent infections' (viral infection followed by bacterial infection) and 'simultaneous infection' (pathogens administered together). Furthermore, we controlled for priming effects by interferon-gamma (IFNγ) treatment before infection and included an LPS control, allowing to identify effects caused by bacterial activation of Toll-like receptor 4 (TLR4).

To simplify the large dataset, and to quantify and score the resulting interactions (Mendeley Data https://data.mendeley.com/datasets/thjzhzdpvc/1, as well as Source Data File), we first introduced simplified metrics for all readouts and then calculated an interaction score for these dynamic parameters that would allow us to capture differences in co-infection dynamics and overall outcome.

As metrics for the three readouts (host cell death, bacterial growth and viral growth) we calculated 1) the area under the curve (AUC), 2) the maximum growth or death, 3) the time of onset of growth

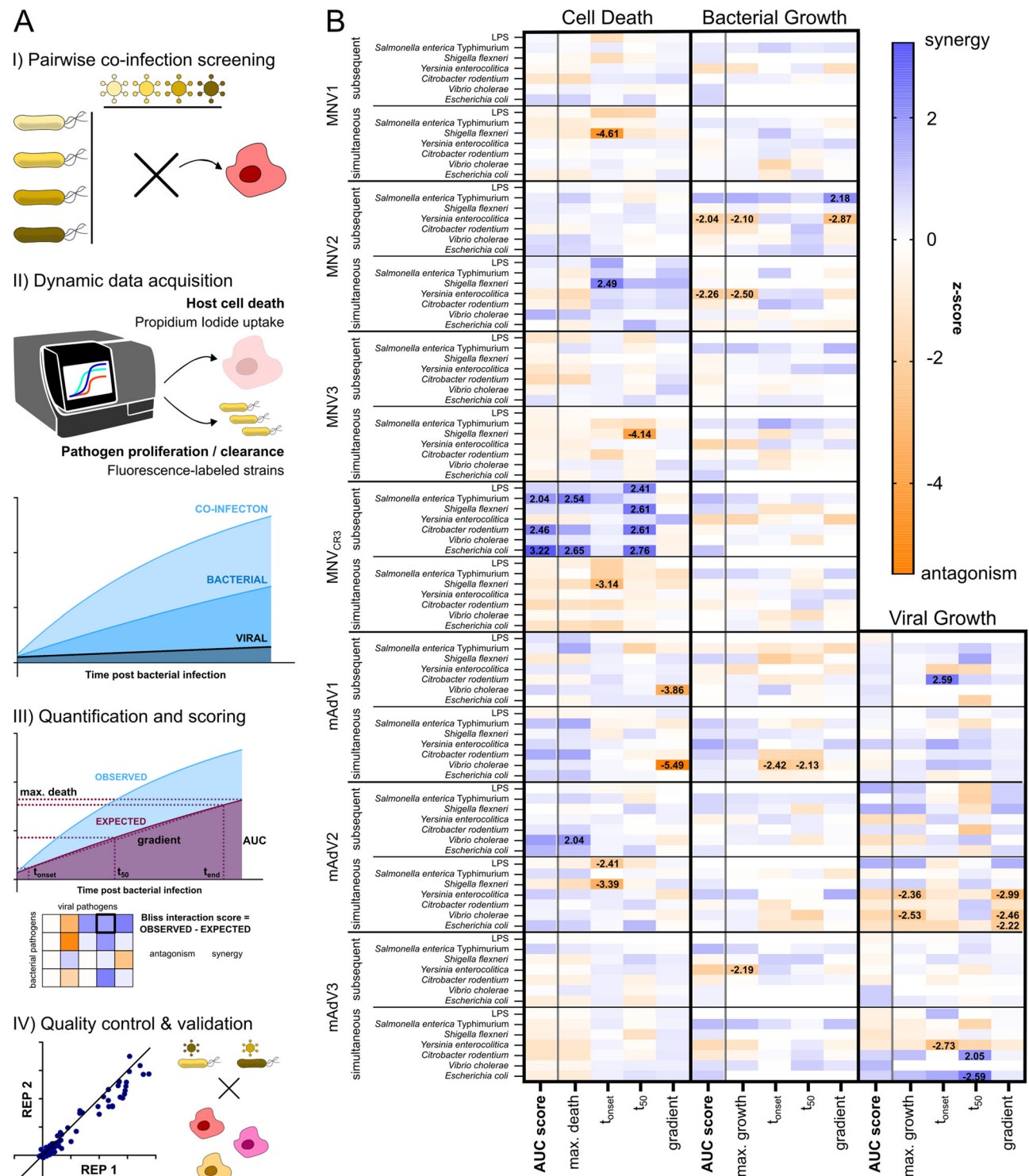

**Fig. 1 | Pipeline for unbiased co-infection screen enables dynamic capture of host-pathogen interactions. A** Workflow to screen interactions during co-infection: I) 6 bacterial enteric pathogens (all Gram-negative, controlled by LPS treatment) and 7 viral strains were employed in pairwise combinatorial co-infections of IFNγ-primed iBMDMs. Both simultaneous co-infection, as well as subsequent co-infection were assessed. II) Using plate-reader-based dynamic fluorescence measurement, host cell death (propidium iodide uptake) and pathogen proliferation or clearance (fluorescently tagged pathogens) were quantified over time for co-infection pairs and individual infections. III) Assuming Bliss neutrality, an expectation for host cell death or pathogen growth (purple curve in the scheme) was

calculated and compared to the observed readout (blue curve). To do so, various dynamic parameters were quantified and assessed. IV) Subsequently to the unbiased screen, replicate reproducibility was evaluated, and a subset of interactions was subjected to validation by orthogonal biochemical assays. **B** Heatmap of z-scores for each pathogen combination (rows) and dynamic metric (columns) for each of the three readouts: host cell death, bacterial growth, viral growth (not measurable for murine norovirus strains due to the lack of a GFP-tag). Synergistic interactions are shown in blue, antagonisms in orange, z-scores larger than 2 and smaller than −2 are indicated by value.

or death, 4) the halftime to maximum ($t_{50}$), as well as 5) the gradient, and normalized them using z-scores (Fig. 1B, Supplementary Data 1). We noticed that the AUC served as a representative metric of the ensemble of dynamic parameters (Figure S2A), as it captured the averages of the scores calculated for each of the three readouts (Figure S2B). Consequently, we assessed replicate reproducibility (Figure S2C) and score distribution (Figure S2D) for the z-scores obtained from the AUC metric. Both quality control measures revealed a sufficient repeatability between biological replicates, as well as a reliable dynamic range in effect size (Figure S2C, D), as indicated by significantly non-zero correlation.

To calculate the interaction score, the expected outcome for each metric was calculated using a Bliss independence model[49], and the observed interaction was compared to the assumed neutral interaction (Fig. 1A, III), i.e. that the two pathogens do not affect or impact each other or the host. In brief, neutral interactions for pathogen growth are presumed to be unchanged to the single infection. For cell death, neutral interactions are defined as follows: The percentage of alive cells during co-infection equals the multiplication of the fraction of alive cells during single infections.

Each combination was assessed in technical triplicate and biological duplicate, and the screening was followed by quality control. Subsequently, a subset of interactions was validated using orthogonal biochemical assays and alternative cell culture models (Fig. 1A, IV).

## Antagonistic interactions between two infecting pathogens are more common than synergies

One goal of the unbiased screen was to evaluate general trends occurring during co-infection of bacterial and viral pathogens. By evaluating the non-normalized Bliss scores, we observed that antagonistic interactions (i.e. a weaker than expected outcome: lower pathogen growth or cell death) were more common than synergies (i.e. a stronger than expected outcome: stronger pathogen growth or cell death) in all three readouts (Fig. 2A, Figure S3A). In the case of host cell death, antagonisms occurred more frequently in simultaneous infections, rather than subsequent infection (Fig. 2A-B), and for pathogen growth, average growth potentials (i.e. the median growth compared to single infection) of 85.9% and 82.4% could be calculated for bacterial and viral proliferation respectively (Fig. 2C). Viral co-infection antagonized the growth of bacterial pathogens, which do not rely on entering host cells, such as *Yersinia*, *Citrobacter* and *Escherichia*, while *Shigella*, and especially *Salmonella*, both of which readily invade host macrophages, displayed more synergistic interactions with respect to bacterial proliferation upon viral co-infection (Fig. 2A, middle panel).

While antagonistic interactions are readily explicable for pathogen proliferation (competition for the same niche, including entry points and mechanisms, resource limitation, activation of different defense pathways), the more frequent occurrence of antagonistic interactions in host cell death seemed counter-intuitive, given the increased clinical severity of co-infections, as compared to single infections. Several studies have however pointed out that cell death and the initiation of inflammation are main contributors to the regulation and restriction of pathogen spread throughout the organism[50–52], hence a reduction of cell death would enable a more systemic dissemination of the pathogen.

Lastly, we interrogated whether the three readouts were correlated in the directionality of the interaction. While we observed a slight positive correlation of host cell death and bacterial growth (Fig. 2D, $p = 0.0036$), we could not establish clear trends for the correlation with the quantified viral growth (Figure S3B, $p > 0.05$). This indicates that, while increased cell death can in part be attributed to bacterial growth, the majority of interactions occurs at specific points and is highly dependent on each pathogen pair and readout. This hypothesis is further strengthened by the overall little overlap in strong interactions co-occurring in different metrics (Fig. 2E), with the largest overlap occurring for cell death and bacterial growth.

## Validation of a subset of interactions underlines the soundness of the resource

To benchmark the quality and reliability of the generated dataset, and in the absence of comparable large-scale studies in the literature, and taking into account the large differences in model systems present in the current literature, we developed a validation strategy, which is based on the use of orthogonal biochemical assays (lactate dehydrogenase (LDH) release to measure cell death and quantification of colony forming units (CFUs) for bacterial proliferation) at adequate time-points. While the validation was performed in RAW264.7, to stay consistent with the cell line used in the screening, we tested alternative cell lines, such as immortalized Bone Marrow Derived Macrophages (iBMDMs), for a subset of pathogen pairs to determine the specificity of the observed interactions. We selected 16 pathogen pairs, spanning all bacterial species and most viral strains that were assessed in the screen. We furthermore chose interaction, so that the largest possible dynamic range of Bliss scores observed in the screen was represented (Fig. 3, black dots), spanning both synergies and antagonisms.

We were able to reproduce the directionality of the interaction (that means replicating synergy or antagonism, irrespective of the strength of the effect) in 81.2% of the tested pairs (Fig. 3A). Notably, antagonisms were more readily replicated than synergistic interactions, and the reliability was correlated with the absolute value of the screening score that was observed (Fig. 3B, C, Figure S4). For host cell death, we were able to replicate all but two tested interactions (Fig. 3B) and could faithfully recapitulate both synergies and antagonisms (see panel insert b) as an example). The directionality observed for bacterial growth could be validated in 12 out of 16 tested pairs (Fig. 3C, see insert c) for example), with *Vibrio cholerae* and pathogenic *Escherichia coli* showing the strongest deviations from the screening score.

Overall, we can conclude that the dataset is reliable and can be validated using alternative methods and readouts. We did so by individual testing interaction pairs using LDH release for cell death and CFU counting for bacterial growth, thereby excluding any artefacts that might arise from the parallelization during the screen, such as plate effects, as well as technical errors due to relying on a single readout. We observed a clear correlation of screening and validation, which was especially true for interaction pairs that deviated more strongly from zero in the screen, however even small deviations from neutrality could be reproduced. Furthermore, the correlation between screening and validation results is more stringent for cell death than for bacterial growth, albeit both of them displaying a positive correlation.

## Murine Adenovirus 3 changes host innate immune responses and alters impact of secondary bacterial infection

In the process of validating the results of the screen using orthogonal biochemical assays and different macrophage cell models, we observed a cell type dependence of the interaction between mAdV3 and *Salmonella*. While the two pathogens displayed a synergy on host cell death for RAW264.7 macrophages (as used in the screening and the validation), this was not the case for iBMDMs (which are another commonly used macrophage model). In fact, we observed an antagonistic interaction that occurred in an IFNγ-priming dependent manner (compare second and third section of Fig. 4A). *Salmonella* is well known to induce cell death through different cell death pathways, among them pyroptosis through the activation of inflammasome complexes[53–55]. One of the main differences between RAW264.7 cells and iBMDMs is the inability of RAW264.7 to activate inflammasomes that require the adaptor protein ASC, since RAW264.7 are deficient for *pycard*, the gene encoding ASC[56]. This suggested an involvement of host innate immune signaling and inflammasome activation upon primary viral infection, which in turn mediated the host cell death response to the secondary *Salmonella* infection.

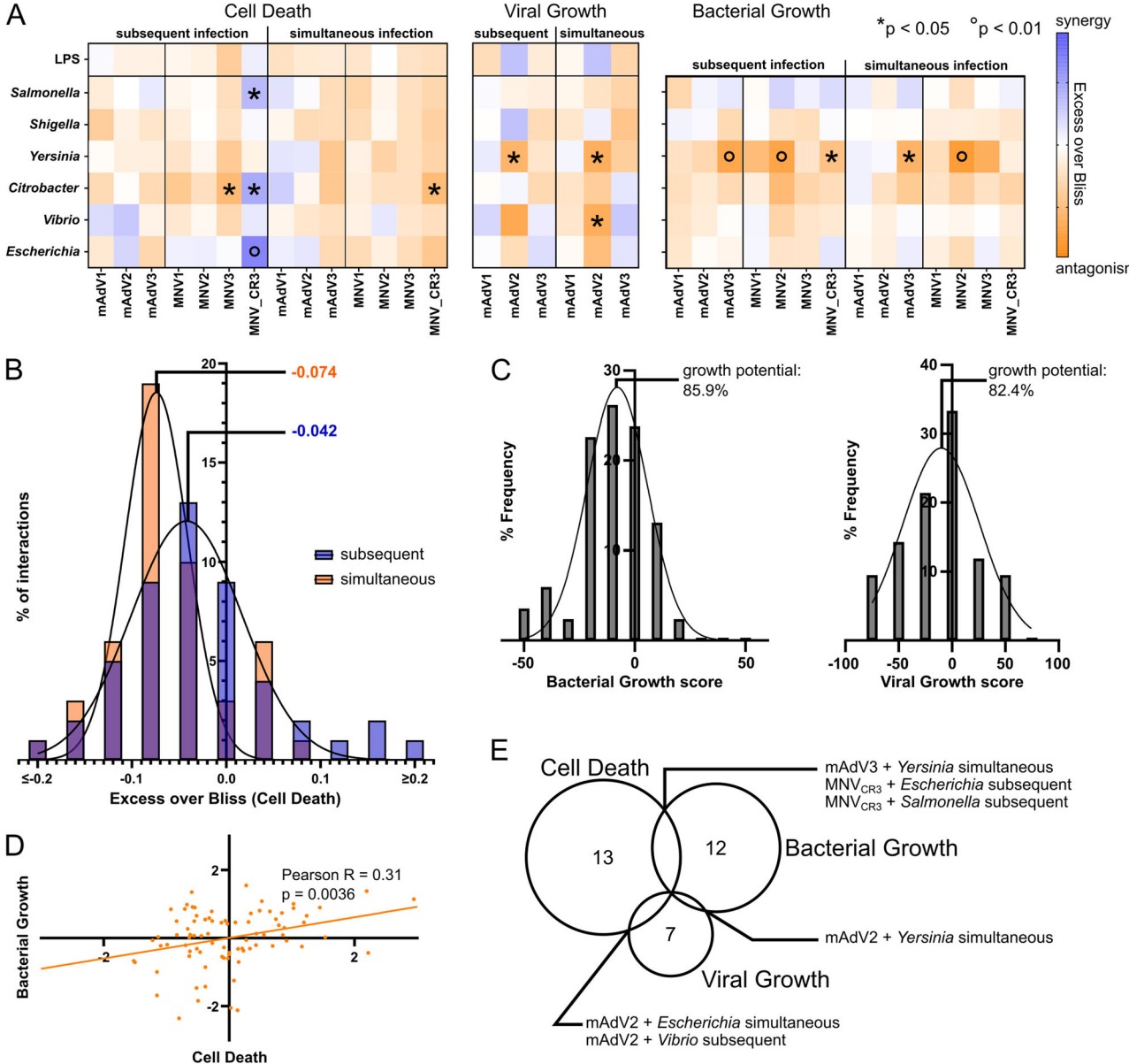

**Fig. 2 | General interaction trends revealed by co-infection screen. A** Separate heatmaps for the three readouts, displaying the non-z-normalized Bliss scores. Bacterial pathogens and LPS control are shown in rows, viral pathogens in columns; each square represents the Bliss score (AUC) for the given interaction. Stars and circles indicate deviations from zero that are significant by the indicated *p*-value (two-sided test, corrected for multiple testing). **B** Histogram of Bliss scores (host cell death) for subsequent (blue) and simultaneous (orange) interactions, including approximative Gaussian of either distribution. The indicated value depicts the median of the Gaussian. **C** Histograms and Gaussian approximation of Bliss scores for bacterial (left) and viral (right) growth. Using the distribution the average growth potential was calculated with respect to the assumption of Bliss independence. **D** Correlation of z-scores for bacterial growth (y-axis) and cell death (x-axis), Pearson R of the linear approximation, and *p*-value for significantly non-zero slope (two-sided test). Each point represents a pathogen pair. E) Number of interactions displaying a z-score larger 1 (synergies) or smaller than −1 (antagonisms) per readout. Pathogen pairs that fall in this category for multiple readouts are indicated in the respective overlap.

To understand the global impact that the primary viral infection with mAdV3 has on the host cell proteome, we performed Stable Isotope Labeling of Amino Acids in Cell Culture (SILAC) upon mAdV3 infection. To do so, we infected iBMDMs with mAdV3 in the presence of heavily labeled ($K^{+8}$, $R^{+10}$) amino acids and compared the newly synthesized proteins with those in an uninfected control, grown in intermediately labeled amino acids ($K^{+4}$, $R^{+6}$). We identified a vast range of proteins involved in cytoskeletal rearrangements being upregulated or induced upon viral infection (Supplementary Data 2, Figure S5A, quality control Figure S5B). Conversely, the most strongly depleted protein was Interferon Inducible Protein 203 (Ifi203), the mouse homolog of IFI16, which is a cytosolic DNA

sensor[57] that plays a role in the response to viral infection[58,59]. Furthermore, IFI16 has been implicated in inflammasome activation via ASC and Caspase-1[60]. Interrogating the most strongly altered host proteins by GO-enrichment revealed an overrepresentation of proteins involved in endocytosis, as expected for the viral infection condition[61], cell cycle regulation, which has been shown to be targeted by Adenoviruses[62], as well as host defense responses and regulation of immune response (Fig. 4B).

As a consequence, we interrogated if mAdV3 is generally able to modify host inflammasome responses by individually activating NLRP3 (by nigericin treatment), AIM2 (by poly-dAdT transfection) and PYRIN (by treatment with UCN-01) and NLRC4 (infection with SPI-1 induced

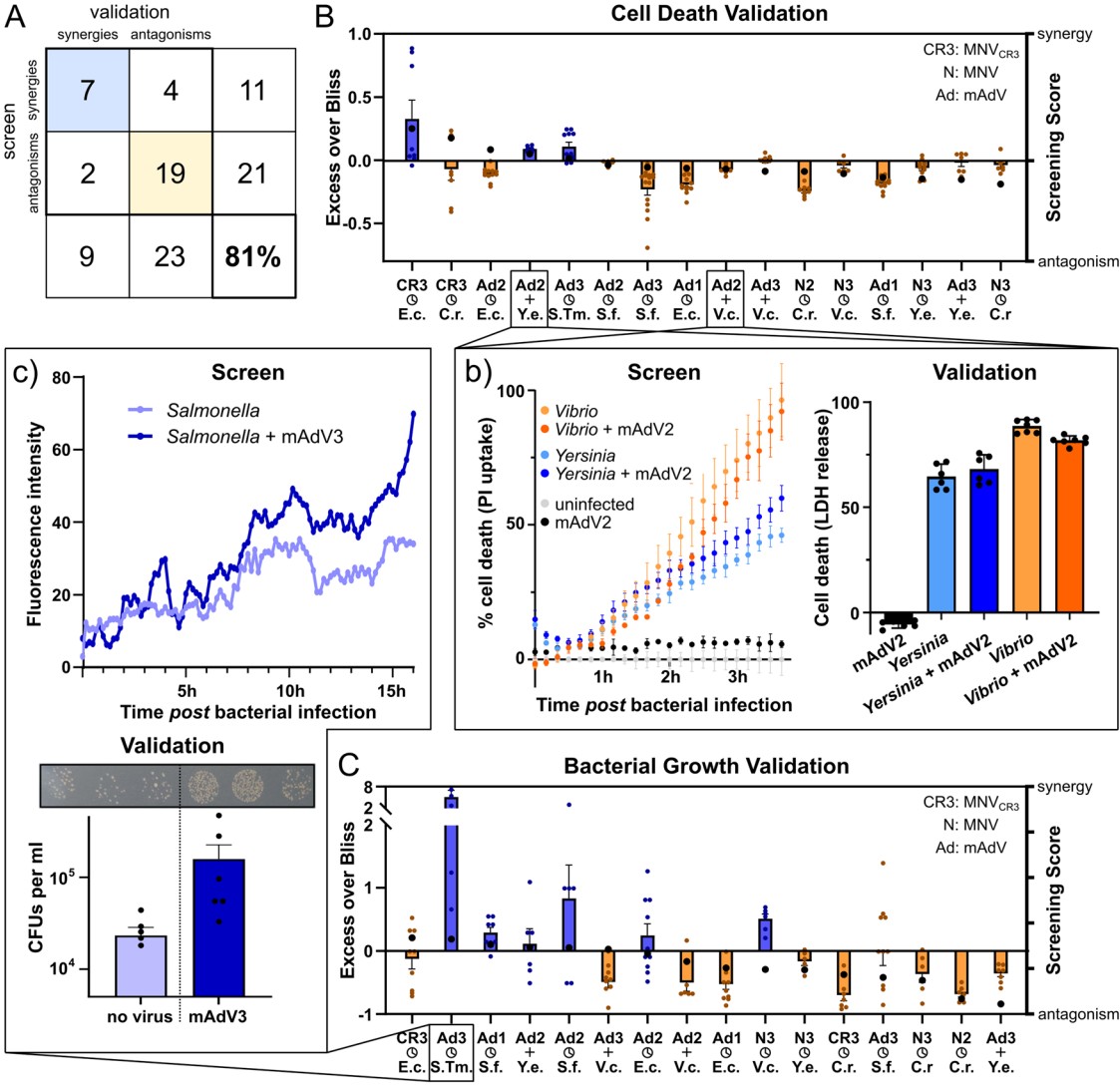

**Fig. 3 | Validation of a subset of interactions by orthogonal biochemical assays.** **A** Matrix displaying the overall recovery of synergies and antagonisms during validation. Interactions were categorized according to their Bliss score (>0: synergies; <0: antagonisms), irrespective of their absolute value. **B** Selected interactions (x-axis), including their screening score (right y-axis, black dots for each interaction), as well as their Bliss score obtained during validation (left y-axis, orange and blue bars, calculated from LDH-release). Bars indicate mean with SEM (based on at least three biological replicates, each with at least two technical replicates). Small insert (b) displays time-resolved fluorescence signal obtained in the screen (left panel, representative of the two biological replicates, error bars indicate standard deviation of three technical replicates) and quantification from LDH-release obtained in the validation (right panel, mean and SEM are indicated, blue bars depict a validated synergy, orange bars a validated antagonism). **C** As for panel B, but concerning bacterial growth as readout. Small insert (c) shows the time-resolved fluorescence signal for the displayed co-infection (proxy for bacterial growth, upper panel, representative replicate from the screening), as well as the validation of this co-infection pair by CFU quantification.

*Salmonella*). Excitingly, while not displaying major changes in cell death (Fig. 4C), mAdV3 infection dampened activation of each ASC-dependent inflammasome, as measured by the secretion of IL-1ß (Fig. 4D) or the formation of ASC specks (Fig. 4E, F). As mAdV3 did not have this impact on the mostly ASC-independent NLRC4 inflammasome, we can conclude that mAdV3 dampens host inflammasome activation at the level of ASC-speck formation, preventing a normal inflammatory response to secondary stimuli. In the case of co-infections (For *Salmonella* see Fig. 4A, for *Shigella* see Figure S5C), this antagonism can lead to a slower host response and an improved survival and proliferation of the bacterial pathogen within its host replicative niche, which is in line with the increased proliferative potential observed in macrophages for *Salmonella* (Fig. 2A, middle panel, and Fig. 3C, insert c)).

## Cytoskeletal remodeling induced by murine Adenoviruses 2 alters infection trajectory of secondary *Yersinia* infection

Of all bacterial pathogens that were assessed in this study, *Yersinia* displayed the strongest antagonistic interactions with co-infecting viruses, showing overall reduced bacterial proliferation or increased clearance (Fig. 2A, middle and right panels). Conversely, murine Adenovirus 2 was heavily antagonized by secondary bacterial infection, highlighting the importance of the host cell as a replicative niche for the viral pathogen (Fig. 2A, middle panel). To understand whether the observed effects on the bacterial and viral pathogens originate from a population effect or were indeed rooted in doubly infected individual host cells, we performed Flow Cytometry (FC) analysis to discern singly and doubly infected populations. Using the fluorescently labeled strains employed in the screening, we were able to determine the

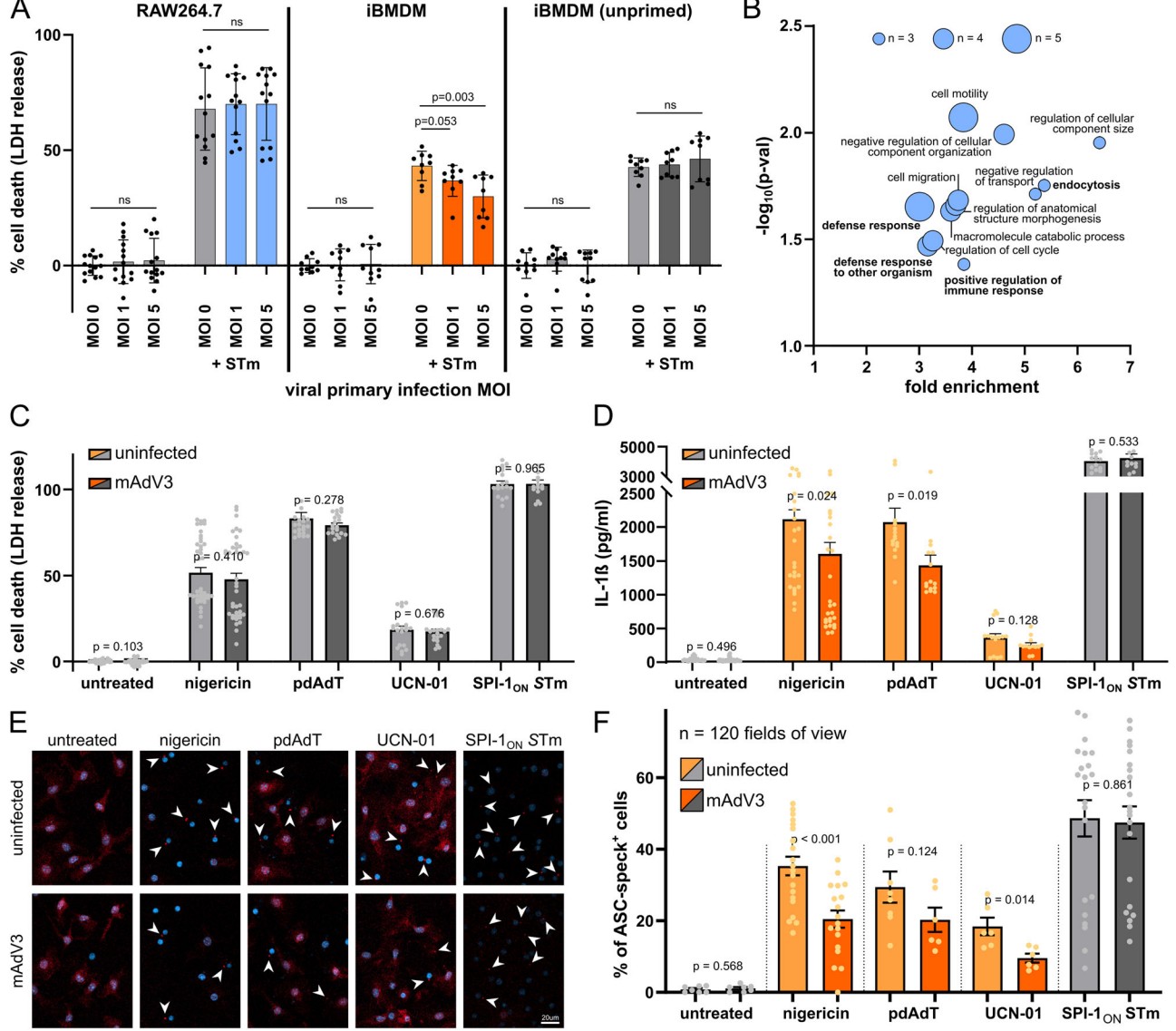

**Fig. 4 | Murine Adenovirus 3 dampens various inflammasome responses in ASC-dependent manner. A** LDH assay in RAW264.7 macrophages, iBMDMs (IFNγ-primed) and iBMDMs (unprimed) after viral pre-infection (for 8 h) with increasing MOIs of mAdV3, as indicated and bacterial secondary infection with STm (right-hand three bars for each cell line) at an MOI of 50 for 16 h. Colors indicate whether co-infection showed a synergistic (blue), antagonistic (orange) or neutral (gray) trend. Each point represents a technical replicate across three biological replicates, bars depict the mean and error bars indicate standard deviation. For statistical assessment, two-sided Student's T-test with Welsh correction was used. **B** GO-term enrichment of proteins with an absolute log$_2$-fold-change larger than 0.5, p value < 0.05. GO-term overrepresentation was performed without correction and only terms with an enrichment >3, at least 3 proteins, and p value < 0.01 (Fisher's exact test, for enrichment only) were selected. Bubble size indicates the number of mAdV3-up- or down-regulated proteins for each term. **C** Cell death as quantified by LDH release for BMDMs that were pre-infected with mAdV3 for 8 h (dark bars) or not (light bars), and subsequently treated with the indicated inflammasome activator for 2 h. p values were obtained by two-sided Student's T-test with Welsh correction and bars represent mean with SEM, n = 254. **D** As in panel C, but with IL−1ß released, measured by ELISA, n = 218. Orange colored bars indicate antagonisms, gray bars represent neutral interactions. **E** Representative fluorescence microscopy images (crop from a field of view acquired with 20x magnification) after staining with ASC-antibody (red) and DAPI (blue) of cells treated as described in panel **C**. Arrows indicate ASC-speck positive cells, scale bar: 20 μm. **F** Quantification of 120 fields of view, represented by the example given in panel E. Coloring and statistics as in panel C, bars represent mean with SEM. For consistency purposes p = 0.0002 is displayed as p < 0.001.

fractions of singly and doubly infected cells for all bacterial pathogens upon mAdV2 primary infection (Figure S6A).

By assuming the independence of the two sequential infections, we could calculate the expected fraction of co-infected cells—which could faithfully be recapitulated for all bacterial pathogens apart from *Yersinia*. Intriguingly, for this specific pathogen combination (mAdV2 and *Yersinia*), we observed an increase in the number of co-infected cells, indicating that a larger fraction of bacteria is taken up by virus pre-infected cells, as compared to uninfected bystanders (Fig. 5A,

quantification: Fig. 5B). In addition to this FC-based assay, we were able to recapitulate the *Yersinia*-specific phenotype on a population level by CFU counting (Figure S6B).

Since *Yersinia* uses its T3SS to evade host phagocytosis[37,63], we next investigated whether depleting the virulence plasmid (pYV-) or deleting specific translocated effector proteins (Δyop), or combinations thereof, alters the fraction of co-infected cells. Interestingly, depletion of the virulence plasmid strongly increased the observed fraction of co-infected cells with respect to the expected fraction.

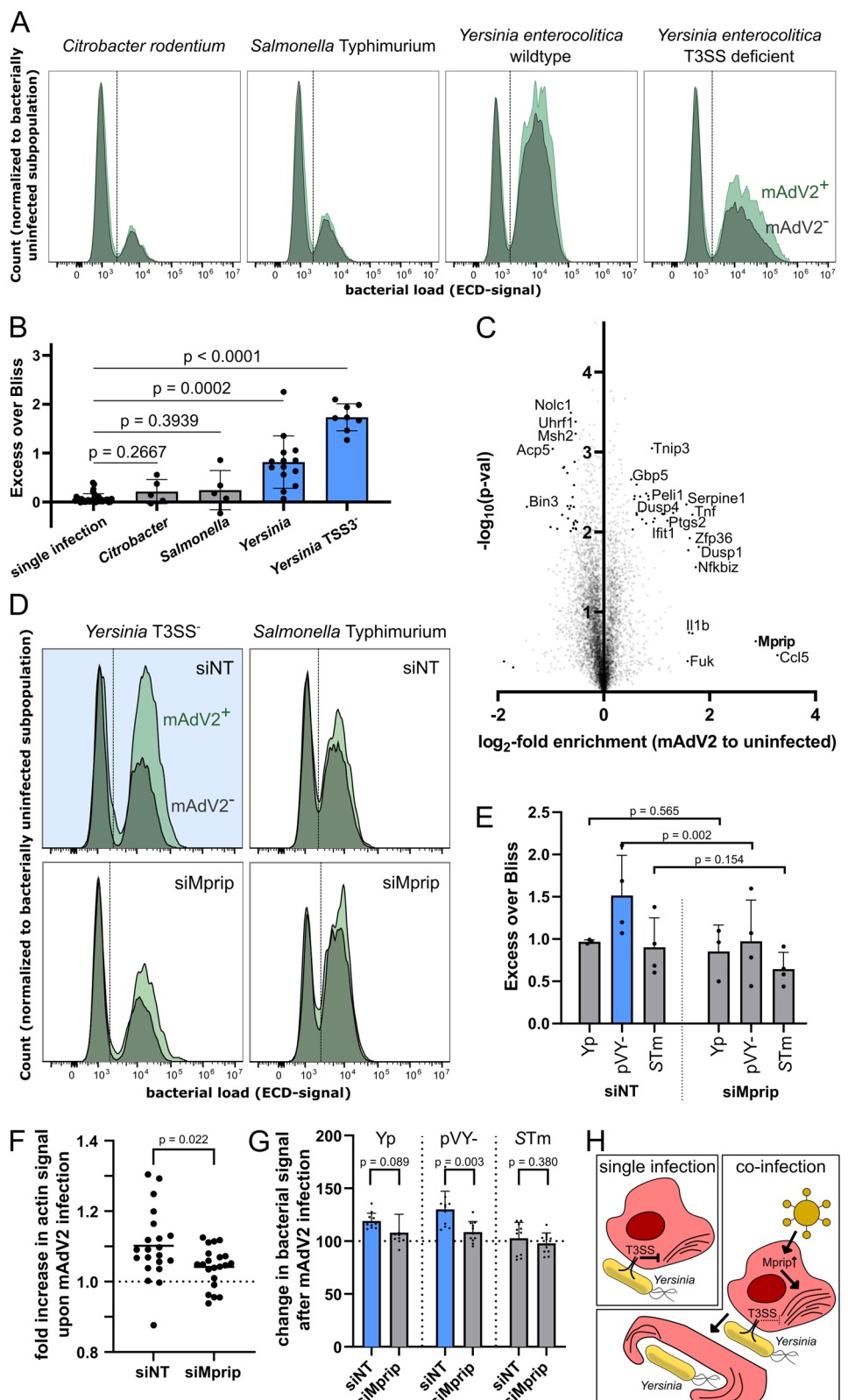

While no single effector deletion or combination of two or three effector proteins could amount to the effect displayed by a T3SS-deficient mutant, effectors YopE and YopH appeared to play the strongest role in the uptake phenotype (Figure S6C).

To further disentangle the interaction between mAdV2 and *Yersinia*, we employed SILAC labeling to identify newly synthesized proteins that were specifically enriched upon viral primary infection (Supplementary Data 3, Fig. 5C, quality control: Figure S6D). One of the most strongly induced proteins was Mprip, which has been implicated in the regulation of actin stress fibers in previous studies[64]; these, in turn play a vital role in cellular motility and phagocytosis[65,66]. We therefore hypothesized that the primary infection with mAdV2, through upregulation of Mprip causes actin remodeling, allowing for the increased uptake of *Yersinia*. To test this, we depleted Mprip-levels

**Fig. 5 | Murine Adenovirus 2 increases host uptake of *Yersinia* in a Mprip-dependent manner. A** Representative distributions of ECD-signal (bacterial load), normalized to uninfected cells (height of the ECD-negative peak), for mAdV2-positive (green) and mAdV2-negative (gray) subpopulations of co-infected cells assessed by FACS. **B** Quantification of Excess over Bliss (observed minus expected fraction of co-infected cells) for mAdV2-coinfections (single infection control: *n* = 34, co-infections: *n* = 32). Each dot represents one biological replicate. Mean and standard deviation are indicated, *p*-values were calculated using two-sided, unpaired Student's T-test with Welsh correction. **C** Enrichment plot obtained from SILAC after mAdV2-infection displaying logarithmic fold-change (mAdV2-infected vs uninfected) and negative logarithmic *p* value (not corrected for multiple testing) of three biological replicates. Only proteins that were found in at least two replicates are shown and selected, highly induced or depleted proteins are indicated by name. Full data, including raw files are available on proteomeXchange (accession number: PXD050625) and in Supplementary Data 3. **D** As in panel A, but after treatment of BMDMs with either a pool of four non-targeting control siRNAs (siNT) or siRNAs against Mprip (siMprip). **E)** Quantification of 3 (*Yp*) or 4 (pVY-, *S*Tm) biological replicates exemplified in panel **D** (i.e. in total, 22 FACS profiles). Excess over Bliss as in panel **B**, *p* values were calculated from paired, two-sided T-test, comparing siNT and siMprip from the same biological replicate. Mean with standard deviation are indicated. **F** For each condition (siNT and siMprip), 21 fields of view (FOV) were quantified (see Methods). Each dot represents the integrated actin intensity, normalized to the overall number of cells (DAPI signal) and divided by the average of the integrated actin intensity in the uninfected control. The *p*-value was calculated by two-sided, unpaired Student's T-test with Welsh correction. **G** 11 FOV were analyzed for each condition. The integrated intensity of the bacterial signal was normalized to the number of cells (DAPI and actin signal) and divided by the average integrated intensity of the virally uninfected sample of the same replicate. Mean with standard deviation is indicated by the bars, *p* values were calculated as in panel **F**. **H** Proposed model for the interaction of mAdV2 and *Yersinia*.

by siRNA knockdown, using a pool of four siRNAs targeting *Mprip* (Figure S6E), and subsequently assessed the uptake of *Yersinia* by the host cells *via* FC. Intriguingly, we were able to revert the T3SS-dependent phenotype we observed for *Yersinia* through Mprip-knockdown, which had no effect on other pathogens, such as *Salmonella* (Fig. 5D, quantification: Fig. 5E), which actively invade the host cell and utilize it as a proliferative niche.

In a next step, we used confocal fluorescence microscopy to visualize the impact of mAdV2-infection on actin remodeling and subsequent uptake (Figure S6F). We observed that the intensity of the actin signal throughout the cell increased upon viral infection, an effect which could be fully reversed upon Mprip knockdown by siRNA (Fig. 5F). By quantifying the bacterial fluorescence signal inside host cells, we were able to show that the bacterial uptake was increased upon viral infection for *Yersinia* (and not *Salmonella*), especially when depleting the virulence plasmid (Fig. 5G). In line with the data obtained through FC, Mprip-knockdown reverted the phenotype, equalizing the uptake for any of the bacterial strains used for infection (Fig. 5G), confirming the role of Mprip in the observed increased uptake of *Yersinia* during co-infection with mAdV2.

In summary, our data suggests that, while during single-pathogen infection with *Yersinia*, where the bacteria actively try to subvert phagocytosis using T3SS effectors, mAdV2 pre-infection induces the production of Mprip, as well as other regulators of phagocytosis, which in turn override the repression exerted by the T3SS effector proteins. Thereby, uptake of *Yersinia* is increased upon mAdV2 pre-infection, especially in the absence of the virulence plasmid and consequently the T3SS (Fig. 5H).

## Discussion

In this study we present a large-scale dataset of pairwise interactions during viral-bacterial co-infection. The data encompasses not only the impact on pathogen clearance or proliferative behavior but spans a bridge to the involvement of the host, its ability to recognize and respond to secondary infections, and intracellular interaction points that mediate the pathogen-pathogen interaction. We strongly believe that this is a valuable resource for the infection biology, host-pathogen interactions and innate immunity research communities, all of which can utilize it as a starting point to form further hypotheses and tether their research to an application in infectious diseases.

In addition, we uncover and describe two vignettes of new biology, representing two distinct ways how viral and bacterial pathogens can interact. By demonstrating that interaction points can fall anywhere within the spectrum from innate immune response, host cell death and inflammation to alterations of the host cytoskeleton and secondary effects on bacterial pathogens, we highlight the complexity of the biological interplay. We further show that pathogen-pathogen interactions can be diverse in dynamic, directionality and mechanism, and therefore warrant further studying.

Starting from a cell-line and priming-dependent antagonism on host cell death, induced by mAdV3 infection, we explored the impact of the viral infection on the ability of the host to activate the inflammatory innate immune signaling pathway known as the inflammasome. Thereby, we uncovered an ASC-dependent dampening effect of the viral infection on inflammasome activatability, which in turn has severe implications for a secondary bacterial pathogen, especially for facultatively intracellular bacteria, such as *Salmonella*. The rewiring of the host innate immune response to persistent viral infection represents a broader means of interaction. Adenoviruses subvert IFNγ-induced genes through their E1A protein[33], and IL-1ß signaling, as well as ASC have previously been shown to play a protective role during Adenovirus infection[67]. Genetic analysis of mAdV3 revealed a closer similarity to mAdV1, rather than other Adenoviruses[33,68], yet we observe differences in interaction patterns with secondary infections. It is therefore conceivable that mAdV3 interrupts innate immune signaling further downstream than IFNγ-mediated induction of gene expression, one hypothesis would be disrupting PYD (Pyrin domain)-PYD-oligomerization, which would explain the observed impact on ASC-dependent inflammasome activation[54].

On a broader scale, further viral pathogens have the ability to modulate host innate immunity[69]. Herpes Simplex Virus 1 (HSV1) degrades host innate immune sensors, such as viperin[70] or IFI16[71] and abrogates cGAS/STING-mediated sensing[72], among others, to evade detection in the early phase of infection. Other viruses that have been implicated in inducing degradation of innate immunity proteins include Pseudorabies virus, targeting IFNAR1[73], or Varicella Zoster Virus, acting on IRF3[74]. Additionally to protein degradation, sequestration[75,76] or post-translational modifications[77,78] are strategies employed by viral pathogens. These effects have not only been described in cell culture models, but have also been shown to display a direct clinical outcome[79], where again, no clear pattern could be deduced: On the one hand SARS-CoV−2 infection can alter patient immune responses, making a subset of recovered patients more susceptible to infection with *Candida albicans*[80], on the other hand, latent HSV-infection confers a protection from subsequent bacterial infection[81].

Aside from host mediation through innate immune signaling, possible interaction points are manifold along the entire route of infection, as are the processes affected by viral pathogens[82–84]. Here, we demonstrate how an upregulation of cytoskeleton reorganizing proteins, which is common in viral infection[85–87], impacts secondary bacterial infection through increased uptake of *Yersinia*. Despite the clear phenotype, which occurred in a species-specific manner, we cannot conclude a general rule for the directionality of interactions mediated through the cytoskeleton (e.g. that viral infection always increases bacterial uptake). We observed different directionalities of interactions in the screen (mAdV2+*Shigella* vs mAdV2+*Yersinia*), indicating that the molecular interactions vary from one pathogen to

**Table 1 | List of bacterial strains and mutants used in this study**

| bacterial strain | mutant | resistance | overnight growth |
|---|---|---|---|
| *Salmonella enterica* Typhimurium SL1344 | Wt, pGG2-dsRed | ampicillin | 37 °C |
| *Shigella flexneri* M90T | Wt, pGG2-dsRed | ampicillin | 37 °C, +Kongo Red (agar plates) |
| *Citrobacter rodentium* ICC169 | Wt, pGG2-dsRed | ampicillin | 37 °C |
| *Vibrio cholerae* 2740-80 | ClpV-mCherry | none | 37 °C |
| *Yersinia enterocolitica* MRS40 | Wt, pGG2-dsRed | ampicillin | 26 °C |
| Pathogenic *Escherichia coli* uti89 | Wt, RFP-expressing | kanamycin | 37 °C |
| *Yersinia pseudotuberculosis* IP2777 | Wt, pFPV-mCherry | ampicillin | 26 °C |
| *Y. pseudotuberculosis* IP2777 | pVY-, pGG2-dsRed | ampicillin | 26 °C |
| *Y. pseudotuberculosis* IP2777 | ΔyopE, pGG2-dsRed | ampicillin | 26 °C |
| *Y. pseudotuberculosis* IP2777 | ΔyopH, pGG2-dsRed | ampicillin | 26 °C |
| *Y. pseudotuberculosis* IP2777 | ΔyopE ΔyopH, pGG2-dsRed | ampicillin | 26 °C |
| *Y. pseudotuberculosis* IP2777 | ΔyopE ΔyopH ΔyopT, pGG2-dsRed | ampicillin | 26 °C |
| *Y. pseudotuberculosis* IP2777 | ΔyopJ, pFPV-mCherry | ampicillin | 26 °C |

**Table 2 | List of viral strains that were used in this study**

| viral strain | genotype | titer | used MOI | co-infection delay |
|---|---|---|---|---|
| Murine Adenovirus (mAdV) 1 | IX-FS2A-GFP | $1.3 \cdot 10^7$/ml | 2 | 6 h |
| Murine Adenovirus (mAdV) 2 | ΔE1A-GFP | $5.0 \cdot 10^6$/ml | 1 | 6 h |
| Murine Adenovirus (mAdV) 3 | IX-FS2A-GFP | $5.0 \cdot 10^7$/ml | 5 | 8 h |
| Murine Norovirus (MNV) 1 | wildtype | $1.3 \cdot 10^7$/ml | 1 | 6 h |
| Murine Norovirus (MNV) 2 | wildtype | $6.3 \cdot 10^6$/ml | 1 | 24 h |
| Murine Norovirus (MNV) 3 | wildtype | $3.0 \cdot 10^7$/ml | 3 | 24 h |
| Murine Norovirus (MNV) CR3 | wildtype | $2.0 \cdot 10^6$/ml | 0.25 | 20 h |

another. Besides, there are published studies describing how invasion of one pathogen excludes infection with another, such as in the case of Norovirus and *Salmonella*[88], *Wolbachia* and Dengue Virus[89], while others indicate a synergistic effect on invasion[90].

While we strongly underline the meaningful impact this research will have for the scientific community, we are conscientiously aware of some of the caveats and shortcomings this study is impacted by. We recognize that this is a large-scale in vitro approach, which focuses on single cell-mediated interactions. In addition to the artificiality of the system, we are not able to comment on systemic effects that one pathogen might have on the other. In the clinical setting, where co-infection concerns a complex organism, rather than a cell model, there is a magnitude of conceivable interaction points, some of which have also been discussed in literature[11,84]. These include systemic depletion or activation of the immune response, paracrine signaling within the tissue or organ, as well as tissue-specific effects that rely on the three-dimensional structure and diverse set of cell types present in a living organism. We are therefore keenly aware that results from the screening cannot be directly translated into an in vivo setting, but require careful further investigation in 3D-cell cultures, such as organoids, or appropriate animal models. Furthermore, this screen is, by means of feasibility, limited in the dimensionality and complexity of possible interactions. Here, we study pairwise viral-bacterial co-infections, yet bacterial-bacterial co-infections, other dynamics or a broader spectrum of assessed pathogens, host systems and other perturbations are imaginable and certainly meaningful. It would furthermore be interesting to include opportunistic pathogens, or microbial communities in a similar approach, both of which have been shown to respond to perturbations by drugs, food-related compounds or other environmental perturbations[91–94]. These caveats, as well as the additional dimensions we were not able to address in this work, highlight the necessity for further research, to which we hope this study incentivizes.

In summary, we present here a unique combination of unbiased screening, assessment of populations, identification of interaction points and mechanistic follow up that reveals new biology. This is, despite its limitations, not only a valuable resource for the community to delve further into the studying of co-infections, including further conditions and overcoming caveats that are inevitably linked to a screening approach, but also a reference for molecular interaction points. We thus seek to raise awareness of the importance of studying co-infections and trigger the identification of host-pathogen-pathogen interaction points which can be targeted and develop new strategies for their disruption.

## Methods

### Ethics statement

The research described in this study complies with all relevant ethical regulations. Primary bone-marrow derived macrophages (BMDMs) were generated using 8 to 12 week old C57CL/6, mice (both male and female) under the guidelines and approval from the Swiss animal protection law (licenses VD3257, Service des Affaires Vétérinaires, Direction Générale de l'Agriculture, de la Viticulture et des Affaires Vétérinaires, état de Vaud). All mice were bred and housed in a specific-pathogen-free facility at $22 \pm 1$ C° room temperature, $55 \pm 10\%$ humidity and a day/night cycle of 12 h/12 h at the University of Lausanne.

### Bacterial and viral strains

All bacterial and viral strains used in this study, alongside their culture conditions and additional details on infection are listed in Table 1 (bacterial strains) and Table 2 (viral strains).

### Cell lines and culture

The following cell models were used throughout this study and, unless specified otherwise, grown at 37 °C, 5% $CO_2$ in the indicated media: RAW264.7 macrophages (ATCC, TIB-71) were cultured in Dulbecco's

Modified Eagle's Medium (DMEM, Gibco), containing 10% Fetal Calf Serum (FCS, Thermo Scientific). BMDMs (harvested from wildtype mice) in DMEM, supplemented with 10% FCS and 20% recombinant Macrophage Colony Stimulating Factor (MCSF, produced in the lab from L929 cells). Immortalized BMDMs (iBMDMs, previously produced in the lab) in DMEM containing 10% FCS and 10% MCSF.

RAW264.7 and iBMDMs were maintained by regular splitting into fresh media in tissue-culture treated flasks (TPP), after scraping and washing cells, and controlling for viability. BMDMs were thawed and seeded into non-tissue culture treated dishes (Falcon) for one day. For seeding, cells were detached by incubation at 4 °C in pre-chilled Phosphate Buffer Saline (PBS, Gibco) for 15 minutes and counted in a Countess automated cell counter with Trypan Blue (Thermo Fisher Scientific) viability stain.

### Screening of host-pathogen-pathogen interactions
The day prior to infection, 30,000 RAW264.7 cells were seeded per well of a 96-well plate and primed for at least 6 h with 10 µg/ml m-IFNγ (Stemcell Technologies). Bacterial overnight cultures (see Table 1) were started by inoculating 3 ml LB-media containing appropriate antibiotics with a single colony from an agar plate, and grown at 30 °C with agitation. For simultaneous co-infection, media was replaced with DMEM containing viral particles at the MOI indicated in Table 2, which was determined prior to the screen to allow for dynamic range by titration and assessment of viral growth and cell death over time.

Cells were placed in a pre-heated centrifuge at 300 G, while the bacterial infection was prepared. Bacterial overnight cultures were adjusted by $OD_{600}$ and washed once in PBS. Assuming a host cell number of 50,000 cells per well, bacteria were added at an MOI of 50 directly onto the viral infection media, as a control, cells were treated with 100 µg/ml Lipo-Polysaccharide B5 (LPS, Invivogen). Cells were centrifuged for 5 minutes at 300 G and placed at 37 °C for 30 minutes. Subsequently, cells were washed once in pre-warmed PBS and the media was replaced with OptiMEM (Gibco) containing 20 µg/ml gentamycin (Invitrogen), 12.5 µg/ml propidium iodide (PI, Thermo Fisher Scientific) and 0.1% TritonX-100 (Tx100, ITW Reagents) where applicable, according to the following layout:

Column 1: Virally uninfected, +PI +Tx100; Columns 2-3: Virally uninfected, media only; Columns 4-6: Virally uninfected, +PI; Columns 7-9: Virus added, +PI; Columns 10-11: Virus added, media only; Column 12: Virus added, +PI +Tx100.

Rows: A: untreated, B: LPS control, C-H: each of the 6 bacterial pathogens.

In the case of subsequent infections, the viral pre-infection was performed by replacing the priming media with viral suspension, and centrifugation for 1 h at 37 °C and 300 G. The cells were maintained at 37 °C as indicated in the strain table, which had been determined before conducting the screen, prior to bacterial infection, which was performed as described above.

For dynamic measurement, plates were placed in a plate reader (Biotek Cytation 5, serial number: 1602037, software version: 03/04/ 2017) and fluorescence intensity was recorded in two channels (1: excitation at 479/20 nm, emission at 520/20 nm; 2: excitation at 550/ 20 nm, emission at 591/20 nm) in 10-minute intervals for at least 16 h at 37 °C, 5% $CO_2$. Due to the lack of GFP-tagging, viral growth could not be assessed fluorometrically for MNV strains, and alternative readout methods, such as plaque-forming unit quantification do neither allow for the necessary throughput nor for the required dynamic (i.e. time-resolved) measurement.

### Data cleanup and primary analysis
Cell death was calculated based on the fluorescence measurements obtained in channel 1. Firstly, the average fluorescence intensity obtained in wells containing DMEM + gentamycin was subtracted for each condition individually to account for background signal of the fluorophores expressed by the bacterial pathogen. Secondly, using the Triton-X100 treated total lysis control (100%) and the uninfected control (0%) as reference points, cell death was quantified for each well. Thirdly, values were averaged across intervals of 30 minutes to reduce measurement artifacts. Finally, values below 0% or above 100% were set to 0% or 100% to maintain biological meaningfulness.

Similarly, pathogen growth was quantified using the intensity values obtained in wells containing DMEM + gentamycin (and hence no PI), using channel 1 for bacterial growth and channel 2 for viral growth, with respect to the uninfected controls. Averaging of values was performed over intervals of 50 minutes to reduce noise.

### Calculation of dynamic metrics and epistatic effects
The expected curve was calculated based on single infections, assuming Bliss independence: For bacterial and viral growth, the comparison occurred directly between singly and doubly infected condition: If bacterial / viral growth occurred to la larger degree in the co-infection, the interaction is deemed a synergy, if the pathogen growth was reduced, an antagonism.

For cell death, the expected curve was calculated for each time-point, as the product of the fraction of live cells quantified in the single infection controls. Dynamic metrics were calculated and defined as follows: max. death: the 98th percentile of all values obtained for cell death; $t_{onset}$: the time when cell death increases to more than 2% of max. death (with respect to the minimum cell death); $t_{50}$: as $t_{onset}$, but for 50% of max. death; gradient: difference between max. death and min. death, divided by the difference between $t_{onset}$ and $t_{end}$ (when 98% of max. death are reached); AUC score: The integral of the curve, based on the average Riemann sum.

In a next step, epistatic effects were calculated as the difference between the observed and the expected value for each metric. Z-transformation was applied to normalize distributions, by dividing the difference to the mean value with the standard deviation across all values for a given metric. Representation as heat maps was done in GraphPad Prism (version 10.2.0) and all raw data, cleaned values and calculated scores are available on Mendeley Data (https://data. mendeley.com/datasets/thjzhzdpvc/1, as well as Source Data File). For the assessment of significance, multiple testing correction was applied.

### Quality control and replicate reproducibility
To evaluate data quality and replicate reproducibility, the two biological replicates were compared by Pearson correlation of each inter-action pair, as well as assessing $p$ values of paired t-tests (see Mendeley Data https://data.mendeley.com/datasets/thjzhzdpvc/1, as well as Source Data File). The second approach was taken to assess whether the two replicates behaved significantly differently. Additionally, the dynamic range was taken into account to determine if a third biological replicate was necessary to ensure data reliability. This was only the case for mAdV3 subsequent infection, for which the average across all three replicates was used for further analysis.

### Lactate dehydrogenase (LDH) assay
To validate detected interactions in cell death, cells were seeded, primed and infected as described above. After the initial bacterial infection, the media was replaced with OptiMEM containing 20 µg/ml gentamycin in all wells and the plate was maintained at 37 °C. At the timepoint that was to be validated, one uninfected and one virally infected well were chosen as total lysis control and Tx100 was added for a final concentration of 0.1% Tx100 for 15 min. 30 µl supernatant per well were added to 30 µl LDH reagent (Sigma Aldrich, prepared as indicated by the manufacturer) for 20 min. The reaction was stopped by adding 30 µl 1 M HCl and the plate was measured on a Biotek plate reader at 490 nm.

Cell death for each well was calculated with respect to the uninfected (0% cell death) and the total lysis (100% cell death) controls. The total lysis control of the virally pre-infected sample was used as quality control to prevent artifacts from excessive cell death and reduction in host cell number prior to bacterial infection.

## Quantification of colony forming units (CFUs)

Parallel to determining host cell death by LDH-release, CFUs were quantified to assess bacterial growth. To do so, all gentamycin-containing OptiMEM was removed, and cells were washed once in pre-warmed PBS. Then, 100µl 0.1% Tx100 were added to lyse host cells and serial 1:5 dilutions of the bacteria-containing lysate were prepared. 7µl of each dilution was spotted on an agar plate containing the appropriate selection antibiotic and grown at the appropriate condition for the respective pathogen. Colonies were counted in technical triplicates and CFUs/ml were quantified with respect to the dilution.

## Calculation of validation rate

To assess the validation rate of the screening approach, we designed the following procedure to determine if interactions could be validated or not: Using LDH release (for cell death) and CFU counting (for bacterial growth) as orthogonal biochemical assays, we probed the directionality (synergistic, i.e. more LDH release / more CFUs, or antagonistic, i.e. less LDH release / fewer CFUs) of each tested interaction. We then compared this result to the Bliss score obtained in the screen. In total, 16 pathogen pairs were tested and assessed in both metrics, thus yielding a set of 32 interactions that were used for validation.

For LDH-release, the expected value was calculated by subtracting the product of the fractions of alive cells in the single infections from 100%. For bacterial growth, the expected value was adjusted to the total host cell number, as approximated by the comparison of the total lysis controls for the virally infected and uninfected samples. The validation Bliss score was then calculated as the difference between the observed and the expected outcome, divided by the expected outcome.

## Stable isotope labeling of amino acids in cell culture (SILAC) during viral infection

To assess the newly synthesized proteins upon viral infection, we employed SILAC labeling, comparing virally infected cells with that of an uninfected population. To do so, iBMDMs were seeded in tissue-culture treated 6-well plates (1 million cells per well) and primed for at least 6 h with IFNγ. Infection with mAdV2 or mAdV3 was performed by adding 750µl viral suspension in iBMDM media containing $R^{+10}$ (U-13C6 99%,15N2 99%, Cambridge Isotope Laboratories) and $K^{+8}$ (U-13C6 99%; U-15N4 99%, Cambridge Isotope Laboratories) (SILAC heavy media), using iBMDM media containing $R^{+6}$ (13C6, 99%, CIL) and $K^{+4}$ (4,4,5,5-D4, 96-98%, CIL) (SILAC intermediate media) as uninfected control. At 16hpi, media was removed, cells were washed twice in PBS and subsequently lysed in 500 µl 100 mM Tris pH7.5, containing 4% SDS (FASP-buffer), supplemented with 10 mM DTT (Merck). Lysates were sonicated to shear DNA, boiled at 95 °C for 5 minutes and cleared by centrifugation at full speed for 10 min. Cleared lysates were further processed at the Protein Analysis Facility of the University of Lausanne.

After determination of protein concentration (tryptophan fluorescence method[95]), H and I samples were mixed at an equimolar ratio (total: 100µg) and digested (SP3 method[96] using magnetic Sera-Mag Speedbeads (Cytiva 45152105050250, 50 mg/ml). To alkylate, proteins were treated with 32 mM iodoacetamide (final concentration) for 45 min at RT in the dark. Precipitation was done on beads (10:1 (w:w) ratio beads:material) using ethanol (final concentration: 60%), and after 3 washes with 80% ethanol, beads were digested in 100 mM ammonium bicarbonate containing 1µg trypsin (Promega #V5073), final volume 50 µl, for 2 h at 37 °C. The same amount of trypsin was added for an additional 1 h of digest. Supernatants were recovered and mixed with two sample volumes of isopropanol containing 1% TFA. Samples were desalted on a strong cation exchange (SCX) plate (Oasis MCX; Waters Corp., Milford, MA) by centrifugation, washed with isopropanol containing 1%TFA, eluted in 200µl 80% MeCN, 19% water, 1% (v/v) ammonia, and dried by centrifugal evaporation.

## Fractionation and liquid chromatography / mass spectrometry (LC/MS)

Samples treated for fractionation and LC/MS as previously published by the Protein Analysis Facility at the University of Lausanne. In brief, samples were fractionated in 6 fractions using the Pierce High pH Reversed-Phase Peptide Fractionation Kit (Thermo Fisher Scientific). The fractions collected were in 7.5, 10, 12.5, 15, 20 and 50% acetonitrile in 0.1% triethylamine (~pH 10), redissolved in 2% acetonitrile with 0.5% TFA and used for LC-MS/MS analysis.

LC-MS/MS analysis was carried out on a Fusion Tribrid Orbitrap mass spectrometer (Thermo Fisher Scientific) connected through a nano-electrospray ion source to an Ultimate 3000 RSLCnano HPLC system (Dionex), via a FAIMS interface. Peptides were separated on a reversed-phase custom packed 45 cm C18 column (75µm ID, 100 Å, Reprosil Pur 1.9µm particles, Dr. Maisch, Germany, 4-90% acetonitrile gradient in 0.1% formic acid (total time 140 min)). Cycling through three compensation voltages (-40, -50, -60V) was used to acquire full MS survey scans at 120'000 resolution. A data-dependent acquisition method in the Xcalibur software (Thermo Fisher Scientific) was set up, optimizing the number of precursors selected ("top speed") of charge 2+ to 5+ from each survey scan, while maintaining a fixed scan cycle of 1 s per FAIMS CV. Peptides were fragmented by higher energy collision dissociation (HCD) with a normalized energy of 32%. The precursor isolation window used was 1.6Th, and the $MS^2$ scans were done in the ion trap. The *m/z* of fragmented precursors was then dynamically excluded from selection during 60 s.

## Proteomic data analysis and GO-term enrichment

As described in other recent publications[97], analysis was performed with MaxQuant 2.1.4.0[98], using the Andromeda search engine[99]. The following modifications were selected: cysteine carbamidomethylation (fixed), methionine oxidation (variable), protein N-terminal acetylation (variable), SILAC heavy labeling ($K^{+8}$ and $R^{+10}$), SILAC intermediate labeling ($K^{+4}$ and $R^{+6}$). The mouse (*Mus musculus*) reference proteome based on the UniProt database (RefProt_Mus_musculus_20230301.fasta, from www.uniprot.org, version of January 2023, containing 55'309 sequences), and a "contaminant" database (most usual environmental contaminants, enzymes used for digestion) were used with a mass tolerance of 4.5ppm on precursors (after recalibration) and 20ppm on MS/MS fragments. All identifications were filtered at 1% false discovery rate (FDR) relative to hits against a decoy database (reversed protein sequences). Filtering and processing of MaxQuant outputs were performed using Perseus (version 1.6.15.0)[100], contaminants were removed, and SILAC ratios were $\log_2$-transformed.

Heavy-to-intermediate-ratios were combined across replicates using the protein name to map across runs. Only proteins which had quantified ratios in at least two replicates were kept and *p*-values were calculated with respect to the null hypothesis that the logarithmic ratio between H and I was 0. For GO-term enrichment, selected proteins (logarithmic fold change larger 0.5, *p*-value smaller than 0.05) were queried for over-representation against the full list of GO terms for biological function (GO Ontology database: https://doi.org/10.5281/zenodo.8436609 Released 2023-10-09) in the Panther database (PANTHER Overrepresentation Test (Released 20231017)), using the full mouse proteome as reference. Fisher's exact test without correction was used for *p*-value analysis and only GO-terms (one per hierarchy group) with at least 3 proteins, a *p*-value smaller than 0.01 and a fold-enrichment larger than 3 were kept.

## Sterile inflammasome stimulation with and without mAdV3 pre-infection

The day prior to the experiment, BMDMs were seeded in 96-well microscopy plates (Greiner) and stimulated the next morning with IFNγ, at least 6 h prior to viral infection with mAdV3 (MOI of 5), which was performed as described above. Cells were left in the incubator overnight and subsequently treated with 100μg/ml LPS, by direct addition to the viral suspension / uninfected control. After 6 h, media was removed and inflammasomes were stimulated by treatment with 5μg/ml nigericin (Sigma-Aldrich) or 12.5μM UCN-01 (Sigma Aldrich), transfection with 125 ng/ml poly-dA-dT (using Lipofectamine LTX, Thermo Fisher Scientific) or infection with *Salmonella* which were subcultured for 3.5 h ($1.4 < OD_{600} < 1.8$) at an MOI of 50. After 2 h, plates were spun down, 100μl supernatant was harvested and used for LDH-release and ELISA for IL-1ß (R&D Systems).

The remaining supernatant was discarded, and the cells were washed twice in PBS. Cells were fixed with 4% paraformaldehyde (PFA, Electron Microscopy Sciences) for 10 minutes at RT and cell membranes were solubilized with 0.1% Tx100, 1% BSA (Thermo Scientific) in PBS. Antibody staining (anti-ASC, Adipogen) was performed at 1:1000 dilution overnight at 4 °C in solubilization buffer and plates were subsequently washed and co-incubated in secondary antibody (Donkey-anti-Rabbit-Alexa-647, Invitrogen, 1:5000) in solubilization buffer at RT for 1 h.

To stain nuclei, Hoechst 33342 (Fisher Scientific) was added for 30 min at a 1:5000 dilution, and cells were washed three times prior to image acquisition. For microscopy, images (z-stack spanning the entire cell) were taken at 20x magnification on a Zeiss LSM800 confocal laser scanning microscope using Zeiss Zen Blue software (version 3.8) with the appropriate laser wavelengths and filters. At least 3 fields of view (z-stacks) were acquired for each biological replicate and ASC-speck-positive cells were quantified by automated counting of total cell number and cells with ASC-specks using ImageJ (version 1.53t), after summing all planes of each z-stack.

ELISA was performed according to the manufacturer's protocol and sample input was diluted where required, according to LDH-release. Plates were read at 450 nm and 570 nm and the concentration of IL-1ß was calculated with respect to a standard curve using the absorption difference between the two channels, spanning 15.625 to 1000 pg/ml (quadratic approximation).

## Analysis of mAdV2-infected populations by FACS

To quantify bacterial uptake, BMDMs were seeded the day prior to viral infection in non-tissue culture treated 24-well plates (Eppendorff) and primed the next morning with IFNγ, at least 6 h before mAdV2 infection at an MOI of 1, overnight. Bacteria were cultured overnight (at 37 °C, *Yersinia* at 26 °C) while shaking and *Yersinia* strains and mutants were subcultured (1:25 dilution) at 26 °C for 1 h and 37 °C for 1 h to induce the T3SS. Bacterial infection at MOI 5 were performed as described above, and the cells were harvested directly after invasion by incubating the infected cells for 15 minutes in chilled PBS at 4 °C. Cells were detached and washed once in PBS, and subsequently stained with violet fixable violet live-dead dye (Thermo Scientific) as described by the manufacturer. Cells were washed once more and fixed in 4% PFA for 10 minutes. Subsequently, cells were transferred into 96-well U-bottom FACS plates (Brand) and flow cytometry was performed on a Cytoflex S (Beckman Coulter), where at least 20000 events were recorded for each condition in each replicate. Cells were gated by forward- and side-scatter (Cells), as well as side-scatter height and area (Single Cells) and by PB450-negativity (live cells). FITC-signal (virus) and ECD-signal (bacteria) were used for the quantification of uninfected, singly, and doubly infected cells, with respect to an uninfected control.

## siRNA-mediated knockdown of Mprip

Pooled siRNAs for Mprip were ordered as SMARTpool from siGENOME which contains the following four siRNAs: GAUCAUCA-GUGGGUGGUUA, GGAAAUGGCAGCGACGAUU, GGAUGGUGGUCG-GAAAGUA, GCAAGUGUCAGAACUGCUU (M-058568-00-0005). Prior to the siRNA transfection, BMDMs were seeded in the appropriate format (50000 cells per well for 96-well plates (microscopy suited), 250000 cells per well for non-tissue culture treated 24-well plates). The transfection mix was prepared by mixing 100μl OptiMEM with 3μl XtremeGENE 9 transfection reagent (Sigma Aldrich) 0.25μM SMART-pool, and left incubating at RT for 15 minutes. 10μl were added to each well of a 96-well plate (50μl for 24-well plates) and cells were grown at 37 °C for 48 h. A pool of 4 non-targeting siRNAs was used as control and transfected using the same procedure.

## Knockdown validation: Polyacrylamide Gel Electrophoresis (PAGE) and Western Blot

Cells were harvested through lysis in FASP-buffer, containing 5% ß-mercaptoethanol and sonicated to shear DNA. Samples were boiled for 5 min at 95 °C, subsequently loaded onto a precast Bis-Tris 4-12% gradient gel (mPAGE, Merck Millipore), and PAGE was performed at 120 V for 1 h. Semi-wet transfer onto Amersham Protran nitrocellulose membrane (Sigma Aldrich) was conducted in a Biorad Trans-Blot Turbo system using transfer buffer (25 mM Tris, 192 mM Glycine, pH8.3 containing 20% methanol): 10 minutes at 1.3 A, 25 V. Membranes were washed in TBS containing 0.1% Tween−20 (Sigma, TBS-T), and subsequently blocked for 1 h in 5% milk in TBS-T while rocking at RT. Rabbit-anti-Mprip (Thermo Scientific) primary antibody was added at 1:1000 dilution and membranes were incubated overnight at 4 °C while rocking. The next day, membranes were washed three times for 5 minutes in TBS-T and subsequently incubated for 1 h in HRP-coupled secondary antibody (goat-anti-rabbit, 1:5000 in milk, SouthernBiotech) or in HRP-coupled anti-Tubulin antibody (Abcam, 1:5000 in milk). After three washes, membranes were incubated in ECL solution (BioRad) and images were acquired on an iBright Imaging System (Thermo Fisher Scientific), using an appropriate exposure time.

## Microscopy image acquisition and quantification after mAdV2-co-infections

Infections, plate preparation and image acquisition were performed as described above. Additionally to Hoechst, cells were stained with phalloidin-Alexa647 (Abcam) for 1 h at a concentration of 1:1000 in PBS. After image acquisition (4 fields of view, in a z-stack spanning at least 7 planes per condition in each replicate), the integrated intensities for DAPI, bacteria and phalloidin were quantified across planes in ImageJ. Subsequently, the phalloidin signal was normalized to DAPI (to quantify increase in actin intensity upon viral infection), and the bacterial signal was normalized to both phalloidin and DAPI (to assess changes in intracellular bacteria).

## Statistical analysis

All analyses, significance testing and visualization was performed in Prism (version 10.2.0), taking into account the necessary prerequisites in test selection, such as multiple testing correction, Welsh correction, (un-)pairedness of the samples and normalization of the values prior to testing. Details of statistical analysis are indicated where required.

## Reporting summary

Further information on research design is available in the Nature Portfolio Reporting Summary linked to this article.

# Data availability

Source data are provided with this paper (see Source Data File), and has been made available in Mendeley Data (https://data.mendeley.com/datasets/thjzhzdpvc/1). This includes the calculation template, as

well as original images. The proteomics data generated in this study have been deposited in the proteomeXchange repository (accession number: PXD050625, https://proteomecentral.proteomexchange.org/cgi/GetDataset?ID=PXD050625). All other data, research material or algorithms used for analysis are available upon request. Source data are provided with this paper.

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

## Acknowledgements
We would like to thank Prof. Dr. Urs Greber and Dr. Silvio Hemmi for providing murine Adenovirus strains, and guidance in pathogen handling, Prof. Dr. Stefan Taube for the murine Norovirus strains, and Prof. Dr. Igor Brodsky for the *Yersinia pseudotuberculosis* mutants that were employed in the study. We would furthermore like to acknowledge EMBO for providing funding to P.W. (ALTF-566–2022), the SNSF for providing funding (TMPFP3_217085 to P.W., 310030_192523 to P.B.) as well as the Federal Council of Switzerland (Swiss Excellence Scholarship) for financially supporting P.W.

## Author contributions
The study was designed by P.W. and P.B. All experiments and analysis were conceptualized and performed by P.W. Initial manuscript writing was done by P.W. and further refined by P.W. and P.B.

## Competing interests
The authors declare no competing interests.
