## [Peer Review File · Nature Communications]

Viral-bacterial co-infections screen *in vitro* reveals molecular processes affecting pathogen proliferation and host cell viabilityREVIEWER COMMENTS

Reviewer #1 (Remarks to the Author):

This study by Walch and Broz develops an in vitro screen to analyze viral-bacterial co-infection dynamics. They apply this screen in RAW264.7 macrophages to seven viral (murine Adenovirus (mAdV) 1, 2 and 3, and murine Norovirus (MNV) 1, 2, 3 and CR3) and six bacterial (Salmonella enterica Typhimurium, Shigella flexneri, Yersinia enterocolitica, Citrobacter rodentium, Vibrio cholerae and pathogenic Escherichia coli) enteric pathogens. They use propidium iodide (PI) staining as a proxy for eukaryotic cell death and measure bacterial and viral growth (for mAdV) using fluorescently labeled strains. This screen was used to classify bacterial-viral pairs as being synergistic, or antagonistic, based on how growth, or cell death, dynamics differed in doubly infected cells compared to expectations derived from single infections. General trends in co-infections relationships were described based on these classifications. They attempted to validate a subset of these interactions in a separate cell line, immortalized Bone Marrow Derived Macrophages (iBMDMs), using alternative methods to quantify cell death (LDH-release) and bacterial growth (CFUs). They then examine two viral-bacterial pairs in more detail (mAdV3 with Salmonella enterica Typhimurium and mAdV2 with Yersinia enterocolitica, providing insights into the mechanisms that may be regulating these interactions.

Major Comments:

1. There is a lack of statistical rigor, particularly in the first half of the manuscript (Figures 1-3, S1, S2, and S4). P-values should be provided in addition to Pearson's R for all scatter plots. This is particularly important for correlations that are highlighted in the text and used to support the author's claims. It would be useful to provide p-values testing if the observed signal is significantly different than expected, including appropriate multiple testing correction. This may be difficult because there are only two biological replicates for the screen, as they opted to increase the number of organisms screened. Perhaps they could narrow their focus to those that had large magnitude effects (e.g., $|z\text{-score}| > 1$, as in figure 2E) that were consistent across biological replicates. These could be used for quantifying general trends in Figure 2 and would have been good candidates for validation experiments.
2. Substantial additional explanation/justification of the screening conditions is needed. What is the tropism for these different viruses? What cell type was actually used in the initial screen (figure legend says iBMDMs but other text seems to suggest RAW cells)? Why were different MOIs and "subsequent" infection timepoints used? What was the "goal" amount of cell death in singly-infected cultures?
3. This assay relies strongly on the estimate of the 'expected' signal derived from single infections. Additional quality control of their approach would be beneficial, this could include a technical control using defined ratios of live/dead host cells, bacteria, and viruses. A 'standard' curve showing accurate quantification of these components and clear disentangling of these signals would be useful. They mention using the "total lysis control (100%) and the uninfected control (0%) as reference points"; additional points would be helpful, and this data should be included as a supplement. The main text should clarify that measurements of pathogen growth were from different wells than those used to measure cell death. Additionally, propidium iodide may not be a specific marker of eukaryotic cell death as it can label all accessible nucleic acids, regardless of origin. This may include nucleic acids from the microbial pathogens, as well as RNA induced by the host in response pathogen exposure.
4. MAdV viral growth appears to be the most reproducible between replicates, while cell death does not appear reproducible. Additional biological replicates would be useful and may be necessary for any conclusions being made from the cell death data.
5. Why was viral growth not assessed for MNV? In the figure legend it says "not measurable", but there are well-established assays that have been used extensively for measuring MNV growth in vitro in RAW or BMDM cultures.
6. An additional explanation of the rationale/validity of using macrophage cell lines (RAW264.7 and iBMDMs) for all bacteria/viruses screened should be presented. They mention that Shigella and Salmonella can invade macrophages, but don't mention why these cell lines are appropriate for the other pathogens tested.
7. The authors repeatedly mention the dynamic range of the assay and use that as an explanation

for the pathogen pairs they tested in their validation experiment. If this was their focus, they should show the correlation between the results of the initial screen and the subsequent validation in iBMDMs. Without showing a test related to this relationship they make the following claims:

- Lines 165-167: "Notably, antagonisms were more readily replicated than synergistic interactions, and the reliability was correlated with the absolute value of the screening score that was observed (Fig. 3B, C)."

- Lines 174-175: "This is especially true the bigger the effect size determined in the screen."

This correlation described in lines 165-167 is not shown or statistically tested. Additionally, the stated relationship is not apparent from the data shown, two of the three most synergistic pairs selected from the screen, MNVCR3 with *Citrobacter rodentium* and mAdV2 with *Escherichia coli* appear to have an antagonistic relationship in the validation experiment. MNVCR3 and *Escherichia coli* had largest effect size in the screen but did not validate, countering the claim made by the authors in lines 174-175.

- Relatedly, they mention that the screen was performed with "viral particles at the MOI indicated in the table of viral strains, which was determined prior to the screen to allow for dynamic range." (Lines 524-525)

The use of different MOIs across viruses and how this may impact the results of their interaction screen should be included in the main text.

8. Figure 3 legend should describe insert (c).

9. There is incongruency in the transition from the results presented in Figure 3 to those presented in Figure 4.

- Line 180-82: "we observed a cell type dependence of the interaction between mAdV3 and *Salmonella*. While the two pathogens displayed a synergy on host cell death for RAW264.7 macrophages, this was not the case for iBMDMs."

Is this data shown somewhere? In Figures 3C and (c), and lines 170-172, this interacting pair was used to highlight consistent results for bacterial growth between the screen using RAW264.7 macrophages presented in Figure 1 and iBMDMs used for validation experiments presented in Figure 3. Additionally, in Figure 3B mAdV3 and *S. Tm.* are used to illustrate a consistent synergistic relationship between the screen and validation experiment in regard to cell death.

10. In lines 88-89 the authors state that "This study thereby presents the community with a reliable, unbiased dataset of pairwise interactions during viral-bacterial co-infection." The authors should include all data from the screen (raw and transformed) as supplementary tables and/or in a stable repository. The summary heatmaps (e.g., Figures 1B and 2A) and scatter plots (e.g., Figures S1 and S2A) are not sufficient if this dataset is being presented as a resource to enable future work.

Minor comments:

1. Please spell out abbreviations in the abstract.

2. Line 23: The authors mention that this work presents "putative targets for new therapeutic approaches." However, they do not highlight or describe therapeutics targets identified in this study anywhere in the manuscript.

3. The two sentences beginning "In addition to HIV..." (line 32) are somewhat confusing, can you please rephrase?

4. It would be helpful to better clarify "synergy" and "antagonism" for each of the assays in terms of expected outcomes (e.g. lower CFUs, more cell death, etc.) as there are not intuitive.

5. Consider removing S1B and adding colored trend lines to S1A.

6. Figure 1A color scale should match the rest of the figures in the manuscript.

7. Figure 1B. Consider showing biological replicates in heatmap.

8. Figure 2B,C – Does fitting gaussian make sense, or is it necessary? The underlying distribution doesn't necessarily look gaussian.

9. Line 180: The "different macrophage cell models" have not been well described in main text at this point. The use of RAW264.7 macrophages for the screen presented in Figure 1 should be included earlier in the manuscript. This information could potentially be added to line 100 by changing the line to "measuring host cell death induction [in RAW264.7 macrophages] by Propidium Iodide..."

10. Lines 386-387: "Bars indicate mean with SEM." Is this of biological or technical replicates?

Reviewer #2 (Remarks to the Author):

Summary:

Here, the interaction between bacterial and viral infection are studied in human macrophages using readouts of pathogen growth, cell toxicity, and pathogen clearance. Consecutive infections were compared to co-infection, and interactions were assessed using the Bliss independence model. A validation was performed using biochemical assays and mouse macrophage cell culture models. Two specific interactions are further described in detail. This work is significant because in clinical medicine, it has long been appreciated that viral infection predisposes to bacterial secondary infection, and this study represents an initial look at the innate immune interactions that occur in this setting, in an in vitro system using a single cell type.

This is an interesting study regarding the level of detail and extensive work performed in the two pairs of Adv and bacterial pathogens, and shows important insights in that specific area. The platform overall is intriguing but I think that the significance of this is over stated, and I think further explanation/justification of methods is needed in many areas.

Abstract

-The abstract describes a "screening", but doesn't describe what the method is or how it was applied; the abstract should explain the methods used and what readouts are measured that support the study conclusions. This is especially important because the title also does not indicate what the method or model used to "dissect" interactions. Space can be made by reducing how much of the abstract is dedicated to describing the significance of the screening method. For example, the last sentence of the abstract describes multiple ways that this work will affect other research (new approaches, new targets, this is a valuable resource, and spark other research); the abstract conclusion should instead be the main finding of the study with a clear connection on its impact (rather than a list of assumed impacts).

-This manuscript is about cell culture assays with one cell type. Human-derived macrophages and mouse-BMDM are not mentioned at all in the title or abstract. Please revise so that these representations of the manuscript are accurate.

Introduction

-co-infections between HIV and other pathogens are described, as the authors mention, but this increased susceptibility to infection is well-studied and is related to both HIV coinfection but also largely due to acquired immunodeficiency. This distinguishes HIV co-infections from the others mentioned; I don't think HIV is a good example here unless the authors can clearly delineate acquired immunodeficiency (ie reduction in T cell function and number) from coinfection that occurs independent of this.

-Please remove results from the introduction (line 72). As a point of style, the last three paragraphs of the introduction are a summary of the results; this should be consolidated into one paragraph.

Methods

-Tables are not numbered, and are not listed in the text.

-Strain details are missing. For example, V. cholerae is listed -what is the serogroup and strain, is this a pathogenic V. cholerae endemic strain (serogroup O1 or O139), or something else? Is the strain listed the parent strain to the mutant shown?

-Line 557: I don't understand what this sentence about "smoothing" is communicating.

Line 561: The Bliss independence model seems to be central to the methods used. Please explain what this is for the reader.

-What is a "validation rate" (line 609) here? It is not clear what this means in the context of the study, and this is not a standard method. Did you design this process for validation of your results? If so please state that and explain rationale for the design, or use references to indicate the source for validity of these methods.

-False discovery rate is not mentioned (other than in proteomics filtering, and there, the acronym is not explained). What was the adjustment for testing multiple hypotheses in these many pairwise

comparisons and readouts? In the statistical analysis (line 779) what does "taking into account the necessary prerequisites in test selection" mean?

-How were readouts chosen (ie why proteomics?) Why were macrophages studied and not other cell types? why were mouse BMDM used for the validation? Please explain rationale for these methods.

Results

-Pathogens chosen have a wide variety of pathogenic mechanisms. For example some of these enteric bacterial pathogens are invasive, others are toxigenic (ie the pathogen is noninvasive and does not come in contact with macrophages). What is the rationale for this choice of pathogens?

-The 81% agreement in Fig 3A is evidence that similar reactions may occur between different species macrophages – this is supportive of the assay reliability, but how do you determine this is "validating" rather than evidence of differences between species cell types?

-The repeated discussion of validation, soundness of resource, reliability, benchmarking etc is not in proportion to the level of validation performed, and lacks an appropriate discussion differences between different species cell types, the singular cell type examined, and the limitations of the experiments. The validation work done does not indicate that this experiment could be repeated as a reliable system with any pathogen pair, and the manuscript is written as if this is generalization. It may be clearer to reframe the paper to introduce the screening platform and validation performed, and focus on the 2 main findings as the primary results of significance.

Reviewer #3 (Remarks to the Author):

Comments to the Author

General Comments:

To understanding how pathogens cause and maintain infection is essential to develop novel therapeutics and prevent outbreaks of emerging disease, while the broadening of accessible methodologies has enabled mechanistic insights into single pathogen infections, the molecular mechanisms underlying co-infections remain elusive in the manuscript titled "Dissecting the impact of enteric viral-bacterial co-infection on the host innate immune response and its implications for pathogenicity". The authors were trying to describe them a first-of-its-kind, parallelized, unbiased screening of pairwise co-infections, ensuring reliability through robust quality control and validation., and subsequently decipher two distinct molecular interaction points with the two stories, first one, mAdV3 modifies ASC-dependent inflammasome responses, altering host cell death and cytokine production, thereby impacting secondary Salmonella infection. Secondly, mAdV2 infection triggers upregulation of Mrip1, a crucial mediator of phagocytosis, which in turn causes increased Yersinia uptake, specifically in virus pre-infected cells. The manuscript does need to be well reorganized and revised, I believe that the manuscript would benefit from copyediting to improve the clarity of the scientific English language writing before publication.

Special points:

Concerns:

1. Considering the complexity of the intestinal environment with various types of intestinal cells, microbiota, and intestinal contents, it should be better to consider, test and analyse using animal models or intestinal organoids to provide in vivo evidences for the two newly discovered mechanisms in co-infection scenarios in the host intestine. Please specify it.
2. The authors in the manuscript is trying to reflect pathogen infection by assessing cell death phenotypes post-co-stimulation. Cell death does not necessarily mean victory for the pathogen. Various forms of cell death, including apoptosis, necroptosis, pyroptosis, and ferroptosis, play important roles not only in clearing damaged or senescent cells but also in inhibiting pathogen spread. The correlation between bacterial growth and cell death in Figure 2D (Pearson R = 0.31) is slight, very probabaly indicating the need to consider additional criteria to support the results.

Please specify it.

Major Essential Revisions :

1. Title: "Dissecting the impact of enteric viral-bacterial co-infection on the host innate immune response and its implications for pathogenicity", I do not see both key results from " the host innate immune response "-focused study and " pathogenicity "-focused study in the manuscript.

The key words " enteric viral-bacterial co-infection", the authors in the manuscript chose the Murine Adenovirus (mAdV) 1,2,3 rather than the enteric virus, to choose the RAW264.7 macrophages and Bone-Marrow-derived macrophages BMDMs rather than the enteric cells, Please specify it.

2. Abstract: Some contents in the former part and last part in the abstract are not necessary to display, and while authors need to focus and highlight the core results with the conclusion, please reorganized and revised it. For example, "Firstly, mAdV3 modifies ASC-dependent inflammasome responses, altering host cell death and cytokine production.....", how to understand mAdV3 modifies ASC-dependent inflammasome responses? how to understand altering host cell death and cytokine production....."?

Minor Essential Revisions, for examples:

1. Some detailed issues: Ensure consistency in the presentation of IL-1b and IL-1 β throughout the text.

2. Line 481: Images corresponding to actin, DAPI, and Yp pVY- in Figur S4F lack color display; supplementary color images are needed.

Reviewer #4 (Remarks to the Author):

General comments

The manuscript is well written with a very compelling argument as to the value of studying co-infections. Some statements need clarification and reassessment of significantly different proteins from proteomics experiment is suggested.

Specific comments

Line 156: 'in the absence of comparable studies in the literature, we developed a validation strategy...' this statement seems overly bold given that co-infection studies have been previously performed and LDH and CFU assays are well established for the purposes described.

Line 161: 'spanning all bacterial species and most viral strains...' this statement is not clear if the authors mean all species within this study or all bacterial species in general (which I doubt is the case).

Line 164: What are explanations for the lack of validation or correlation upon reproducing the directionality of the interactions? How do the authors conclude that the approach is reliable with 12/16 bacterial species correlating?

Line 173: what are the alternative methods and readouts?

Line 250: Was siRNA knockdown expression confirmed? Yes, in Fig. S4 for one gene – what about other three genes indicated within the main text?

Line 265: As Salmonella also uses a T3SS, are there parallels observed between the dataset with Yersinia?

Line 604, 625: modify as to not start the sentence with a number (unless written in full)

Line 669: for infection samples, the relevant pathogen databases should also be searched to determine if any peptides map to both mouse and pathogen for removal upon host proteome analysis.

Line 678: for determination of significantly different proteins, multiple hypothesis testing (e.g., Benjamini-Hochberg) for an FDR-corrected p-value should be performed for robustness and to remove potential false positives.

Line 803: the PRIDE accession number, along with reviewer login information should be provided.

References: add italics to scientific names

REVIEWER COMMENTS

Reviewer #1 (Remarks to the Author):

This study by Walch and Broz develops an in vitro screen to analyze viral-bacterial co-infection dynamics. They apply this screen in RAW264.7 macrophages to seven viral (murine Adenovirus (mAdV) 1, 2 and 3, and murine Norovirus (MNV) 1, 2, 3 and CR3) and six bacterial (Salmonella enterica Typhimurium, Shigella flexneri, Yersinia enterocolitica, Citrobacter rodentium, Vibrio cholerae and pathogenic Escherichia coli) enteric pathogens. They use propidium iodide (PI) staining as a proxy for eukaryotic cell death and measure bacterial and viral growth (for mAdV) using fluorescently labeled strains. This screen was used to classify bacterial-viral pairs as being synergistic, or antagonistic, based on how growth, or cell death, dynamics differed in doubly infected cells compared to expectations derived from single infections. General trends in co-infections relationships were described based on these classifications. They attempted to validate a subset of these interactions in a separate cell line, immortalized Bone Marrow Derived Macrophages (iBMDMs), using alternative methods to quantify cell death (LDH-release) and bacterial growth (CFUs). They then examine two viral-bacterial pairs in more detail (mAdV3 with Salmonella enterica Typhimurium and mAdV2 with Yersinia enterocolitica, providing insights into the mechanisms that may be regulating these interactions.

We would like to express our gratefulness for the time and effort you invested in reviewing the manuscript and for the constructive input and suggestions you provided. We apologize for any unclarities that the original manuscript contained, and we have dedicated significant time and effort to improve the manuscript on different aspects. We hope that these changes, which are reflected in this revised version, meet your standard and expectations.

Major Comments:

1. There is a lack of statistical rigor, particularly in the first half of the manuscript (Figures 1-3, S1,S2, and S4). P-values should be provided in addition to Pearson's R for all scatter plots. This is particularly important for correlations that are highlighted in the text and used to support the author's claims. It would be useful to provide p-values testing if the observed signal is significantly different than expected, including appropriate multiple testing correction. This may be difficult because there are only two biological replicates for the screen, as they opted to increase the number of organisms screened. Perhaps they could narrow their focus to those that had large magnitude effects (e.g., |z-score| > 1, as in figure 2E) that were consistent across biological replicates. These could be used for quantifying general trends in Figure 2 and would have been good candidates for validation experiments.

We thank the reviewer for the level of detail with which they read the manuscript and for the suggestions on how to improve statistical analysis of our datasets. To address the raised concerns in the revised manuscript, we included p-values in the respective figure panels, and assessed whether correlations were significantly different from zero (see Figures 1B, 2A, 2D, S2A-C, S3, S4, S6B-D of the revised manuscript). We furthermore added indication of significant interactions in the heatmaps. For all statistical testing, we employed multiple testing correction using ANOVA. With respect to the quantification of general trends, we would like to highlight that Fig. 2B-D and Fig. S2 display a quantification of these trends, and we included further clarification in the text.

2. Substantial additional explanation/justification of the screening conditions is needed. What is the tropism for these different viruses? What cell type was actually used in the initial screen (figure legend says iBMDMs but other text seems to suggest RAW cells)? Why were different MOIs and “subsequent” infection timepoints used? What was the “goal” amount of cell death in singly-infected cultures?

We would like to apologize for not adding enough clarity and depth to our descriptions. In the revised manuscript, we have added substantial background information on the host cell model, the selected pathogens and the general motivation to the introduction section and the results section. We have also restructured the results part by including a new section describing the single pathogen infections which we had performed in RAW cells preparation of the co-infection screen. Of note, single infection were included in the original submission but not described separately.

The single infections (Figure S1) show that all pathogens chosen can infect RAW cells, replicate to different extent and induce variable levels of host cell death, thus confirming that macrophages are a valid experimental model. We furthermore clarified and expanded the related methodology sections. The single infection optimization also allowed us to choose the appropriate MOI, as well as the adequate infection timepoint for the ‘super-infection’ setup.. While we did not have a specific „goal” amount, we intended to: 1) keep the conditions as consistent as possible across pathogens, while 2) allowing for both increases and decreases in cell death and pathogen growth during co-infection, so that both antagonisms and synergies could be determined. Therefore, in the viral titration depicted in Figure S1G, the MOI that is shown in bold was chosen over the other tested conditions.

3. This assay relies strongly on the estimate of the ‘expected’ signal derived from single infections. Additional quality control of their approach would be beneficial, this could include a technical control using defined ratios of live/dead host cells, bacteria, and viruses. A ‘standard’ curve showing accurate quantification of these components and clear disentangling of these signals would be useful. They mention using the “total lysis control (100%)”

and the uninfected control (0%) as reference points"; additional points would be helpful, and this data should be included as a supplement. The main text should clarify that measurements of pathogen growth were from different wells than those used to measure cell death. Additionally, propidium iodide may not be a specific marker of eukaryotic cell death as it can label all accessible nucleic acids, regardless of origin. This may include nucleic acids from the microbial pathogens, as well as RNA induced by the host in response to pathogen exposure.

Propidium iodide uptake is a very widely used technique to quantify cell membrane permeabilization, and the standard practice in the field is to use "total lysis control (100%) and the untreated/uninfected control (0%) to define reference points. Hence, we did not perform linearity assessment for the readout itself (by, as you suggested, mixing different ratios of alive and dead cells). We did, however, perform a titration of viral infection during the setup phase of the screen. This was meant to optimize infection conditions such as timing or MOI, and also relates to your previous point about the rationale behind the different delays in the subsequent infections. Figure S1G depicts these titration curves (for the three murine Adenoviruses) quantifying cell death and viral proliferation over time.

Regarding the specificity of PI and its use as a cell death marker:

It is correct that PI can interact with both DNA and RNA of the host cell, but since it only intercalates into double-helical regions of both DNA and RNA, the RNA signal is usually negligible compared to the DNA signal. Notably, mRNA should not result in much PI signal. Furthermore, when using PI as a marker for cell death, it is not important what type of nucleic acid is bound, since the goal is only to distinguish dead cells that become PI positive from live cells that do not acquire PI staining as they have an intact membrane. Finally, it is unclear how much nucleic acids from the pathogen itself would contribute to the PI signal, but we expect it to be minimal for the following reasons: 1) PI only stains bacteria that have died and lost membrane integrity, thus most of the intracellular bacteria would remain unstained as host cell death does not automatically result in bacterial killing. 2) The DNA content of a bacterial cell is much lower than the amount of host DNA (approx. 5×10^6 base pairs in *Salmonella* genome vs. 3.3×10^9 bp in human genome)

4. MAdV viral growth appears to be the most reproducible between replicates, while cell death does not appear reproducible. Additional biological replicates would be useful and may be necessary for any conclusions being made from the cell death data.

We thank you for pointing out that limitation. We acknowledge that Cell Death shows the highest variability across the different assessed readouts, yet nonetheless, the replicate correlation is present ($R = 0.38$) and significantly different from zero ($p < 0.01$), as indicated in Figure S2C. Furthermore, we

hope that our validation by LDH release provides additional confidence into the conclusions we make.

5. Why was viral growth not assessed for MNV? In the figure legend it says “not measurable”, but there are well-established assays that have been used extensively for measuring MNV growth in vitro in RAW or BMDM cultures.

The reason is that we do not have access to fluorescently tagged MNV strains. It is correct that plaque-forming units (PFU) counts could be used as an alternative way of assessing MNV growth in cells, however this approach doesn't allow us to do the high-throughput dynamic measurements necessary for the screen. In the revised manuscript we now thus specifically mention that the MNV strains employed in this study are not GFP tagged (in the legend to Figure 1B), and that other described quantification techniques, such as the quantification of plaque-forming units neither allow for the necessary throughput, nor a dynamic (i.e. time-resolved) measurement (specified in the methods section).

6. An additional explanation of the rationale/validity of using macrophage cell lines (RAW264.7 and iBMDMs) for all bacteria/viruses screened should be presented. They mention that Shigella and Salmonella can invade macrophages, but don't mention why these cell lines are appropriate for the other pathogens tested.

We have incorporated this suggestion in the introduction, where we added a substantial piece of detailed explanation of our rationale. In brief, macrophages are a valuable cell model, since they can be infected and respond to all pathogens used in the study, being the ‘first-responder’ of the innate immune system (see Figure S1). Other cell types, while scientifically highly interesting, are not necessarily permissive to all infectious agents or don't display the same properties with respect to cellular innate immune response. Moreover, despite the different virulence strategies employed, macrophages play a key role in coordinating the immunity against all of the chosen enteric bacteria and were shown to interact with or respond to these bacteria *in vivo*, in particular after disruption of the epithelial cell barrier:

- *Salmonella*: PMID: 25533091
- *Shigella*: PMID: 10874731
- *Yersinia*: PMID: 14500510, PMID: 9841926
- *Citrobacter*: PMID: 24043764, PMID: 28612839
- *Vibrio*: PMID: 37295405, PMID: 33042146
- *E. coli*: PMID: 27766279, PMID: 21802164

7. The authors repeatedly mention the dynamic range of the assay and use that as an explanation for the pathogen pairs they tested in their validation experiment. If this was their focus, they should show the correlation between the results of the initial screen and the subsequent validation in iBMDMs.

Without showing a test related to this relationship they make the following claims: Lines 165-167: "Notably, antagonisms were more readily replicated than synergistic interactions, and the reliability was correlated with the absolute value of the screening score that was observed (Fig. 3B, C)." Lines 174-175: "This is especially true the bigger the effect size determined in the screen."

This correlation described in lines 165-167 is not shown or statistically tested. Additionally, the stated relationship is not apparent from the data shown, two of the three most synergistic pairs selected from the screen, MNVCR3 with *Citrobacter rodentium* and mAdV2 with *Escherichia coli* appear to have an antagonistic relationship in the validation experiment. MNVCR3 and *Escherichia coli* had largest effect size in the screen but did not validate, countering the claim made by the authors in lines 174-175. Relatedly, they mention that the screen was performed with "viral particles at the MOI indicated in the table of viral strains, which was determined prior to the screen to allow for dynamic range." (Lines 524-525). The use of different MOIs across viruses and how this may impact the results of their interaction screen should be included in the main text.

To address this comment, we performed correlation analysis of the screening and the validation, and have included a new supplementary figure (Fig. S4) to display the results of this analysis. We were able to show that the screen and validation indeed correlate ($R=0.54$ for Cell Death and $R=0.46$ for Bacterial Growth), and that this correlation is significantly different from zero. We therefore hope to have provided the statistical basis to make the claim that effect size in the screen is predictive of the performance in the validation. We furthermore hope that our response to point 3, where we cite the viral titration curve prior to the screen provides the necessary basis for our claims with respect to allowing for dynamic range.

8. Figure 3 legend should describe insert (c).

We have added a description of the insert (c) in the figure legend.

9. There is incongruency in the transition from the results presented in Figure 3 to those presented in Figure 4. Line 180-82: "we observed a cell type dependence of the interaction between mAdV3 and *Salmonella*. While the two pathogens displayed a synergy on host cell death for RAW264.7 macrophages, this was not the case for iBMDMs." Is this data shown somewhere? In Figures 3C and (c), and lines 170-172, this interacting pair was used to highlight consistent results for bacterial growth between the screen using RAW264.7 macrophages presented in Figure 1 and iBMDMs used for validation experiments presented in Figure 3. Additionally, in Figure 3B mAdV3 and *S. Tm.* are used to illustrate a consistent synergistic relationship between the screen and validation experiment in regard to cell death.

This result is shown in Figure 4A, and we changed the phrasing of the text to state this more clearly. The synergistic effect with respect to Cell Death in RAW264.7 is very small, yet consistent (as shown in Figure 3B – these experiments were carried out in RAW264.7), however, as indicated by the orange bars in Figure 4A, we noticed an antagonistic effect of mAdV3-STm co-infection in iBMDMs. This effect was also dependent on the priming of the macrophages prior to infection, which led us to investigate the impact of the virus on iBMDMs, as well as the difference between the two cell lines. We hope that the clarifications we introduced in the revised manuscript suffice to provide better understanding of our observations and rationale for follow-up.

10. In lines 88-89 the authors state that “This study thereby presents the community with a reliable, unbiased dataset of pairwise interactions during viral-bacterial co-infection.” The authors should include all data from the screen (raw and transformed) as supplementary tables and/or in a stable repository. The summary heatmaps (e.g., Figures 1B and 2A) and scatter plots (e.g., Figures S1 and S2A) are not sufficient if this dataset is being presented as a resource to enable future work.

We realize that the tone with which we describe the implications of the dataset might have struck a wrong note and have rephrased the respective sections in the manuscript. We have also ensured that the full dataset, including all raw data, processed data and analysis pipelines are publicly accessible (for both the screening data and the proteomics). We had already done so for the initial submission, providing a draft version to be accessed, but due to a technical error on our side, it appears that this access was not possible for the reviewers – we would like to apologize for this:

- PRIDE repository (PXD050625)
- Mendeley Data (<https://data.mendeley.com/datasets/thjzhzdpvc/1>)

To simplify the data accessibility, we have furthermore included three additional supplementary tables that contain the screening data (Table S1), as well as the SILAC proteomics data (Tables S2 and S3)

Minor comments:

1. Please spell out abbreviations in the abstract.

We have adapted the text accordingly. We would however prefer to keep the commonly used short protein names ASC and Mprp instead of ‘Apoptosis-associated speck-like protein containing a caspase recruitment domain’ and ‘Myosin Phosphatase Rho Interacting Protein’ in the abstract due to better readability. In the main text, we specify the full names during the first mentioning of the respective protein.

2. Line 23: The authors mention that this work presents “putative targets for new therapeutic approaches.” However, they do not highlight or describe therapeutics targets identified in this study anywhere in the manuscript.

We have removed that statement, and have generally tried to adapt the phrasing to better reflect the scope of the study.

3. The two sentences beginning “In addition to HIV...” (line 32) are somewhat confusing, can you please rephrase?

We have changed the phrasing in the revised manuscript and hope that this provides more clarity.

4. It would be helpful to better clarify “synergy” and “antagonism” for each of the assays in terms of expected outcomes (e.g. lower CFUs, more cell death, etc.) as there are not intuitive.

We realize that the characteristics of interaction directionalities are not intuitive to all readers and have therefore added this information in the revised manuscript.

5. Consider removing S1B and adding colored trend lines to S1A.

We believe that keeping S1A and S1B separate adds clarity and improves readability, since panel S1A is already fairly crowded, and the overlay of multiple colored trendlines might create more confusion.

6. Figure 1A color scale should match the rest of the figures in the manuscript.

We have adapted the color scale, so that it matches the other figures in the manuscript.

7. Figure 1B. Consider showing biological replicates in heatmap.

Since we have deposited all original data, including the individual biological and technical replicate values, in an online repository (and have since ensured that it is accessible), we hope that those who want to assess the data in more depth can do so in a more comprehensive way than studying a larger heatmap.

8. Figure 2B,C – Does fitting gaussian make sense, or is it necessary? The underlying distribution doesn’t necessarily look gaussian.

To avoid overfitting, we started with the initial assumption of Gaussian distribution. Given that the R^2 values are 0.97, 0.98, 0.96, 0.86 for the four different Gaussian fits depicted in Figure 2B and 2C, we remain confident that a Gaussian fit is appropriate. Nonetheless, if you – or any other interested reader would like to apply a different statistical method to our data, we ensured the availability on Mendeley data (<https://data.mendeley.com/datasets/thjzhzdpvc/1>)

9. Line 180: The “different macrophage cell models” have not been well described in main text at this point. The use of RAW264.7 macrophages for the screen presented in Figure 1 should be included earlier in the manuscript. This

information could potentially be added to line 100 by changing the line to “measuring host cell death induction [in RAW264.7 macrophages] by Propidium Iodide...”

The revision of the introduction should now include this information at an earlier, and therefore more appropriate point.

10. Lines 386-387: “Bars indicate mean with SEM.” Is this of biological or technical replicates?

We have specified the origin of the data in the text.

Reviewer #2 (Remarks to the Author):

Summary:

Here, the interaction between bacterial and viral infection are studied in human macrophages using readouts of pathogen growth, cell toxicity, and pathogen clearance. Consecutive infections were compared to co-infection, and interactions were assessed using the Bliss independence model. A validation was performed using biochemical assays and mouse macrophage cell culture models. Two specific interactions are further described in detail. This work is significant because in clinical medicine, it has long been appreciated that viral infection predisposes to bacterial secondary infection, and this study represents an initial look at the innate immune interactions that occur in this setting, in an in vitro system using a single cell type.

This is an interesting study regarding the level of detail and extensive work performed in the two pairs of Adv and bacterial pathogens, and shows important insights in that specific area. The platform overall is intriguing but I think that the significance of this is over stated, and I think further explanation/justification of methods is needed in many areas.

We appreciate the constructive feedback that stems from thorough and critical reading – thank you for investing the time and effort to improve our research. We have added additional evidence to support our claims, and have at the same time clarified the scope of the study. In case our tone came across as overstating, we would like to apologize and hope that this reviewed manuscript meets your expectations.

Abstract

-The abstract describes a “screening”, but doesn’t describe what the method is or how it was applied; the abstract should explain the methods used and what readouts are measured that support the study conclusions. This is especially important because the title also does not indicate what the method or model used to “dissect” interactions. Space can be made by reducing how much of the abstract is dedicated to describing the significance of the screening method. For example, the last sentence of the abstract describes multiple ways that this work will affect other research (new approaches, new targets, this is a valuable resource, and spark other research); the abstract conclusion should instead be the main finding of the study with a clear connection on its impact (rather than a list of assumed impacts).

We are very thankful for your comment and suggestion. In the revised manuscript, we have tried to address your concerns by more clearly stating the scope and experimental design of the study in the abstract (and throughout the manuscript).

-This manuscript is about cell culture assays with one cell type. Human-derived macrophages and mouse-BMDM are not mentioned at all in the title or

abstract. Please revise so that these representations of the manuscript are accurate.

Related to the prior point, we have revised the abstract to mention the model in more detail and hope that this will more accurately capture the representation of the manuscript.

Introduction

-co-infections between HIV and other pathogens are described, as the authors mention, but this increased susceptibility to infection is well-studied and is related to both HIV coinfection but also largely due to acquired immunodeficiency. This distinguishes HIV co-infections from the others mentioned; I don't think HIV is a good example here unless the authors can clearly delineate acquired immunodeficiency (ie reduction in T cell function and number) from coinfection that occurs independent of this.

We agree that co-infections in the context of HIV are mechanistically very different from the type of interactions that we are trying to uncover in this study. We apologize for not having made that distinction clearer. To address this, the revised manuscript now includes a paragraph introducing the different levels that interactions during co-infection can occur: on the systemic level, in paracrine signaling, or on a single cell level. We have furthermore included more evidence on co-infections in the context of Influenza, which has been studied on the cellular level to a larger extent.

-Please remove results from the introduction (line 72). As a point of style, the last three paragraphs of the introduction are a summary of the results; this should be consolidated into one paragraph.

Thank you, we adapted the text accordingly.

Methods

-Tables are not numbered, and are not listed in the text.

We have labeled the two tables (bacterial and viral strains) accordingly, and cite them in the respective sections.

-Strain details are missing. For example, *V. cholerae* is listed -what is the serogroup and strain, is this a pathogenic *V. cholerae* endemic strain (serogroup O1 or O139), or something else? Is the strain listed the parent strain to the mutant shown?

We apologize for not having specified the strain details in the original version of the manuscript. We have now done so. To address your two specific points:

- The strain of *Vibrio cholerae* used has previously been described in a study by Prof. John Mekalanos (Harvard University) (DOI: <https://doi.org/10.1016/j.cell.2013.01.042>);
- for all mutants used, the wildtype is the parental strain.

-Line 557: I don't understand what this sentence about "smoothing" is communicating.

We apologize for the confusion and have changed the text accordingly to provide more clarity. In brief, the idea behind the „smoothing“ is using averaging across a certain time interval to reduce technical noise by the plate reader.

Line 561: The Bliss independence model seems to be central to the methods used. Please explain what this is for the reader.

We apologize that this hasn't been clear in the original version – we have now tried to add more detail and explanation around the concept of Bliss independence, which is, as you very rightfully point out, central to the screening approach.

-What is a "validation rate" (line 609) here? It is not clear what this means in the context of the study, and this is not a standard method. Did you design this process for validation of your results? If so please state that and explain rationale for the design, or use references to indicate the source for validity of these methods.

The validation strategy was designed by us, following approaches taken by other screening studies. We have now added more detail in the methodology section and sought to clarify the approach in the results section to give a clearer motivation for the chosen approach.

-False discovery rate is not mentioned (other than in proteomics filtering, and there, the acronym is not explained). What was the adjustment for testing multiple hypotheses in these many pairwise comparisons and readouts? In the statistical analysis (line 779) what does "taking into account the necessary prerequisites in test selection" mean?

We have changed and amended the text accordingly and hope that these further specifications provide clarity about the stringency of statistical analysis we sought to apply.

-How were readouts chosen (ie why proteomics?) Why were macrophages studied and not other cell types? why were mouse BMDM used for the validation? Please explain rationale for these methods.

We apologize for the lack of clarity in the initial version of the manuscript. In this revised version, we have added more detail with respect to our rationale for the experimental system(s) we used, as well as more detail to clarify these details – these details were not only added to the Methodology section, where appropriate, but also to the results section. In brief, macrophages are a valuable cell model, since they can be infected and respond to all pathogens used in the study (see Figure S1), other cell types, while scientifically highly interesting, are not necessarily permissive to all infectious agents or don't display the same properties with respect to cellular innate immune response. The validation was mostly performed in RAW264.7 cells (as now more clearly

stated in the revised manuscript). However, to assess generalizability of our screening to other macrophage cell types, we also tested iBMDMs in several interactions and, in the case of mAdV3-infection, identified cell-type specificities which we could in a subsequent step trace back to the competence of the different cells to induce inflammasome activation.

Results

-Pathogens chosen have a wide variety of pathogenic mechanisms. For example some of these enteric bacterial pathogens are invasive, others are toxigenic (ie the pathogen is noninvasive and does not come in contact with macrophages). What is the rationale for this choice of pathogens?

We have added a paragraph that addresses this question in the introduction. Our goal was to assess the impact of viral co-infections on a wide number of enteric bacteria with the goal to identify general as well as pathogen-specific effects. We thus selected bacteria with similar as well as distinct virulence strategies, e.g. invasive or toxigenic. Moreover, despite the different virulence strategies employed, macrophages play a key role in coordinating the immunity against all of the chosen enteric bacteria and were shown to interact with or respond to these bacteria in vivo, in particular after disruption of the epithelial cell barrier:

- *Salmonella*: PMID: 25533091
- *Shigella*: PMID: 10874731
- *Yersinia*: PMID: 14500510, PMID: 9841926
- *Citrobacter*: PMID: 24043764, PMID: 28612839
- *Vibrio*: PMID: 37295405, PMID: 33042146
- *E. coli*: PMID: 27766279, PMID: 21802164

-The 81% agreement in Fig 3A is evidence that similar reactions may occur between different species macrophages – this is supportive of the assay reliability, but how do you determine this is “validating” rather than evidence of differences between species cell types?

We have added more clarification to our validation strategy and hope that this resolves the questions you are raising. In brief, the validation as presented in Figure 3 was performed in RAW264.7 macrophages (so the same cell line as the screen), but using different readouts and a smaller scale. In large scale approaches, where many conditions are tested in parallel, technical artifacts can arise from plate effects, the small scale of the reaction vessel or the parallelized quantification. By assessing individual interactions one-by-one using different techniques we sought to provide methodological validity. With respect to the difference in cell lines, we agree – very likely there are cell type or -line specific effects, as shown in Figure 4A and discussed thereafter. This is why the presented validation was performed in RAW264.7.

-The repeated discussion of validation, soundness of resource, reliability, benchmarking etc is not in proportion to the level of validation performed, and lacks an appropriate discussion differences between different species cell types, the singular cell type examined, and the limitations of the experiments. The validation work done does not indicate that this experiment could be repeated as a reliable system with any pathogen pair, and the manuscript is written as if this is generalization. It may be clearer to reframe the paper to introduce the screening platform and validation performed, and focus on the 2 main findings as the primary results of significance.

We thank you for the critical reading and the time you invested in your feedback. We apologize if our phrasing came across as overstating and have tried to clarify and specify the scope of the project, e.g. in the abstract, the introduction, the results and the discussion section.

Reviewer #3 (Remarks to the Author):

Comments to the Author

General Comments:

To understanding how pathogens cause and maintain infection is essential to develop novel therapeutics and prevent outbreaks of emerging disease, while the broadening of accessible methodologies has enabled mechanistic insights into single pathogen infections, the molecular mechanisms underlying co-infections remain elusive in the manuscript titled "Dissecting the impact of enteric viral-bacterial co-infection on the host innate immune response and its implications for pathogenicity". The authors were trying to describe them a first-of-its-kind, parallelized, unbiased screening of pairwise co-infections, ensuring reliability through robust quality control and validation., and subsequently decipher two distinct molecular interaction points with the two stories, first one, mAdV3 modifies ASC-dependent inflammasome responses, altering host cell death and cytokine production, thereby impacting secondary Salmonella infection. Secondly, mAdV2 infection triggers upregulation of Mpr1p, a crucial mediator of phagocytosis, which in turn causes increased Yersinia uptake, specifically in virus pre-infected cells.

The manuscript does need to be well reorganized and revised, I believe that the manuscript would benefit from copyediting to improve the clarity of the scientific English language writing before publication.

We are sorry to hear that you haven't found our research and data convincing. In this updated manuscript, we have tried to address your comments to the best of our abilities and have invested significant effort into restructuring the manuscript, adding further details to the significance testing, as well as reformulating the text to better reflect the scope of the study.

Special points:

Concerns:

1. Considering the complexity of the intestinal environment with various types of intestinal cells, microbiota, and intestinal contents, it should be better to consider, test and analyse using animal models or intestinal organoids to provide in vivo evidences for the two newly discovered mechanisms in co-infection scenarios in the host intestine. Please specify it.

We appreciate your question regarding the biological meaningfulness of the model system we established. We fully agree that interactions between pathogens

can occur at various levels and are multifactorial and -dimensional. The scope of the study is to identify molecular interaction points for two pathogens, as well as the role that the host plays in mediating this interactions. An *in vivo* model, while of course being more physiologically relevant, is too complex to clearly distinguish interactions on different levels and identify molecular interaction points. Furthermore, neither an *in vivo* screen, nor an organoid-based system would allow for the throughput needed to study the breadth of pathogen combinations that we sought out to assess.

In this revised manuscript, we have tried to clarify the scope and the claims of this study and have spelled out the rationale of using an *in vitro* model (throughput, accessibility, reduction of complexity) in more detail. We have furthermore added a clearer statement that results from this study cannot directly be taken as fact *in vivo* and discuss the requirement to assess interactions in a more complex and physiological system. However, we see this as a separate step that is not within the scope of this project.

2. The authors in the manuscript is trying to reflect pathogen infection by assessing cell death phenotypes post-co-stimulation. Cell death does not necessarily mean victory for the pathogen. Various forms of cell death, including apoptosis, necroptosis, pyroptosis, and ferroptosis, play important roles not only in clearing damaged or senescent cells but also in inhibiting pathogen spread. The correlation between bacterial growth and cell death in Figure 2D (Pearson R = 0.31) is slight, very probabaly indicating the need to consider additional criteria to support the results. Please specify it.

We agree that cell death does not imply beneficiality or detriment for the pathogen (as a matter of fact, pathogens carefully balance pro- and anti-inflammatory signaling in the host), and we did not claim so in the manuscript. We have furthermore discussed the role of cell death (including its protective effect) throughout the manuscript. With respect to the point raised about the correlation of cell death and bacterial growth: This result was not used to postulate that increased cell death means advantageous growth conditions for the pathogen. In the revised manuscript, we have further clarified that our hypothesis on this correlation is that increased bacterial growth leads to increased cell death, since the host responds to the larger presence of pathogens by a stronger induction of cell death.

Major Essential Revisions :

1. Title: "Dissecting the impact of enteric viral-bacterial co-infection on the host innate immune response and its implications for pathogenicity", I do not see both key results from " the host innate immune response "-focused study and " pathogenicity "-focused study in the manuscript. The key words " enteric viral-bacterial co-infection", the authors in the manuscript chose the Murine Adenovirus (mAdV) 1,2,3 rather than the enteric virus, to choose the

RAW264.7 macrophages and Bone-Marrow-derived macrophages BMDMs rather than the enteric cells, Please specify it.

We have changed the title to better reflect the scope and the rationale of the study.

2. Abstract: Some contents in the former part and last part in the abstract are not necessary to display, and while authors need to focus and highlight the core results with the conclusion, please reorganized and revised it. For example, "Firstly, mAdV3 modifies ASC-dependent inflammasome responses, altering host cell death and cytokine production.....", how to understand mAdV3 modifies ASC-dependent inflammasome responses? how to understand altering host cell death and cytokine production....."?

We have updated the abstract and have added more details on the conceptualization and the core results.

Minor Essential Revisions, for examples:

1. Some detailed issues: Ensure consistency in the presentation of IL-1b and IL-1 β throughout the text.

We have updated and harmonized the representation of IL-1 β .

2. Line 481: Images corresponding to actin, DAPI, and Yp pVY- in Figur S4F lack color display; supplementary color images are needed.

We thank you for this comment and would like to point you to the repository of original data (including all microscopy images), which we have made available: Mendeley (<https://data.mendeley.com/datasets/thjzhzdpvc/1>). To enhance visibility and contrast, we would prefer to keep the images in grayscale, but we have added a remark that all files can be found on Mendeley Data.

Reviewer #4 (Remarks to the Author):

General comments

The manuscript is well written with a very compelling argument as to the value of studying co-infections. Some statements need clarification and reassessment of significantly different proteins from proteomics experiment is suggested.

We thank the reviewer for this very positive feedback, we have tried to address sections of the manuscript that required clarification, and have since also strengthened the statistical assessment of the claims we make by including additional analyses, statistical testing and stringency.

Specific comments

Line 156: 'in the absence of comparable studies in the literature, we developed a validation strategy...' this statement seems overly bold given that co-infection studies have been previously performed and LDH and CFU assays are well established for the purposes described.

We apologize for striking the wrong tone in this statement – we do by no means want to discredit previous studies. We have therefore changed the phrasing and added context to this statement.

Line 161: 'spanning all bacterial species and most viral strains...' this statement is not clear if the authors mean all species within this study or all bacterial species in general (which I doubt is the case).

We have now clarified that we mean strains that were assessed in the screen.

Line 164: What are explanations for the lack of validation or correlation upon reproducing the directionality of the interactions? How do the authors conclude that the approach is reliable with 12/16 bacterial species correlating?

We have added additional correlation analyses to the validation approach (Figure S4), as suggested and we also added further details to better explain our validation strategy and quantification.

Line 173: what are the alternative methods and readouts?

We have specified this in the text now.

Line 250: Was siRNA knockdown expression confirmed? Yes, in Fig. S4 for one gene – what about other three genes indicated within the main text?

We are unsure which three other genes you refer to in your question. We used a pool of four siRNAs, all targeting the same host gene, Mrip (we clarified this in the text now). Other than Mrip, we did not perform any other targeted knockdowns.

Line 265: As *Salmonella* also uses a T3SS, are there parallels observed between the dataset with *Yersinia*?

This is a very interesting question (generally comparing pathogens to each other). Indeed, we see general patterns between „professional invaders“, such as *Salmonella* and *Shigella* – since *Yersinia* is using its T3SS to prevent uptake by host macrophages, we actually see a very different pattern (as also discussed in the mechanistic follow-up related to Figure 5).

Line 604, 625: modify as to not start the sentence with a number (unless written in full)

We have changed the manuscript accordingly.

Line 669: for infection samples, the relevant pathogen databases should also be searched to determine if any peptides map to both mouse and pathogen for removal upon host proteome analysis.

In this specific experiment, we wanted to focus on changes to the host proteome, rather than the viral pathogen. While the raw counts are published online alongside (in a PRIDE repository: PXD050625), so that these questions could be assessed. Since viral infection takes time to establish, the quantity of viral proteins that are newly synthesized might however be quite low. Furthermore, the PRIDE repository contains all relevant information regarding the measurement, peptide search and analysis parameters that were employed in mass spectrometry.

Line 678: for determination of significantly different proteins, multiple hypothesis testing (e.g., Benjamini-Hochberg) for an FDR-corrected p-value should be performed for robustness and to remove potential false positives.

We apologize for not having clearly stated where multiple testing correction was performed – we have done so in the revised manuscript. For the SILAC data in an infection context, the dynamic range is fairly restricted due to the low abundance of newly synthesized proteins within the short timeframe of infection. Therefore, the multiple testing strategies that we assessed did not converge, and we have hence depicted the non-adjusted p-values. In case other investigators want to apply more stringent methods or threshold to the data, we encourage them to do so by accessing the Mendeley Data repository (<https://data.mendeley.com/datasets/thjzhzdpvc/1>) or the PRIDE repository (PXD050625), where all raw and unprocessed data can be found.

Line 803: the PRIDE accession number, along with reviewer login information should be provided.

We would like to apologize strongly for the inconvenience the inaccessibility of the data caused. We have now ensured that the full dataset, including all raw data, processed data and analysis pipelines are publicly accessible (for both the screening data and the proteomics). We had already done so for the initial

submission, providing a draft version to be accessed, but due to a technical error on our side, it appears that this access was not possible for the reviewers – we would like to apologize for this.

References: add italics to scientific names

We have now formatted the references by italicizing species names. The automatic Nature-style formatting option of the citation software we employed did not have seemed to capture this.

REVIEWERS' COMMENTS

Reviewer #1 (Remarks to the Author):

The authors have reasonably addressed my prior concerns by adding further rationale and clarification throughout the manuscript, as well as toning down conclusions.

Reviewer #2 (Remarks to the Author):

The authors have addressed my concerns.

Reviewer #3 (Remarks to the Author):

Pairwise screening of viral-bacterial co-infections in vitro reveals molecular interaction points that affect pathogen proliferation and host cell viability

Summary: This study with the title "Pairwise screening of viral-bacterial co-infections in vitro reveals molecular interaction points that affect pathogen proliferation and host cell viability" by Walch and Broz tried to develop an in vitro screen to analyze viral-bacterial co-infection dynamics and to reveals molecular interaction points that affect pathogen proliferation and host cell viability. They apply this screen in RAW264.7 macrophages to seven viral (murine Adenovirus (mAdV) 1, 2 and 3, and murine Norovirus (MNV) 1, 2, 3 and CR3) and six bacterial (Salmonella enterica Typhimurium, Shigella flexneri, Yersinia enterocolitica, Citrobacter rodentium, Vibrio cholerae and pathogenic Escherichia coli) enteric pathogens. This screen was used to classify bacterial-viral pairs as being synergistic, or antagonistic, based on how growth, or cell death, dynamics differed in doubly infected cells compared to expectations derived from single infections. General trends in co-infections relationships were described based on these classifications. They attempted to validate a subset of these interactions in a separate cell line, immortalized Bone Marrow Derived Macrophages (iBMDMs), using alternative methods to quantify cell death (LDH-release) and bacterial growth (CFUs). They They then examine two viral-bacterial pairs in more detail (mAdV3 with Salmonella enterica Typhimurium and mAdV2 with Yersinia enterocolitica, providing insights into the mechanisms that may be regulating these interactions.

Concerns:

1. The new title "Pairwise screening of viral-bacterial co-infections in vitro reveals molecular interaction points that affect pathogen proliferation and host cell viability" changed a lot compared to the last version title "Dissecting the impact of enteric viral-bacterial co-infection on the host innate immune response and its implications for pathogenicity", the revised title is broad, I may have concern it is not suitable for the results and conclusion in the revised manuscript, and the authors should specify seven viral (murine Adenovirus (mAdV) 1, 2 and 3, and murine Norovirus (MNV) 1, 2, 3 and CR3) and six bacterial (Salmonella enterica Typhimurium, Shigella flexneri, Yersinia enterocolitica, Citrobacter rodentium, Vibrio cholerae and pathogenic Escherichia coli) enteric pathogens and two cells (RAW264.7 macrophages and iBMDMs immortalized Bone Marrow Derived Macrophages) in the manuscript with the focus (molecular interaction points).
2. Considering the complexity of the intestinal environment with various types of intestinal cells, microbiota, and intestinal contents, based on the results described in vitro, it should be better to discuss, to consider, test and analyse using animal models or intestinal organoids to provide in vivo evidences for the two newly discovered mechanisms in coinfection scenarios in the host intestine.

Reviewer #4 (Remarks to the Author):

The authors have thoughtfully addressed the points I raised; however, I do not agree that an FDR-corrected p-value for the SILAC data can be justifiably excluded from the analysis and left for

other researchers to explore. Given the nature of the work and the importance of statistical robustness for accurate and reliable interpretation of the proteomics datasets, an FDR (e.g., 5%) should be set.

Response to the reviewers' comments

Reviewer #3 (Remarks to the Author):

Pairwise screening of viral-bacterial co-infections in vitro reveals molecular interaction points that affect pathogen proliferation and host cell viability

Summary: This study with the title “Pairwise screening of viral-bacterial co-infections in vitro reveals molecular interaction points that affect pathogen proliferation and host cell viability” by Walch and Broz tried to develop an in vitro screen to analyze viral-bacterial co-infection dynamics and to reveals molecular interaction points that affect pathogen proliferation and host cell viability. They apply this screen in RAW264.7 macrophages to seven viral (murine Adenovirus (mAdV) 1, 2 and 3, and murine Norovirus (MNV) 1, 2, 3 and CR3) and six bacterial (Salmonella enterica Typhimurium, Shigella flexneri, Yersinia enterocolitica, Citrobacter rodentium, Vibrio cholerae and pathogenic Escherichia coli) enteric pathogens. This screen was used to classify bacterial-viral pairs as being synergistic, or antagonistic, based on how growth, or cell death, dynamics differed in doubly infected cells compared to expectations derived from single infections. General trends in co-infections relationships were described based on these classifications. They attempted to validate a subset of these interactions in a separate cell line, immortalized Bone Marrow Derived Macrophages (iBMDMs), using alternative methods to quantify cell death (LDH-release) and bacterial growth (CFUs). They They then examine two viral-bacterial pairs in more detail (mAdV3 with Salmonella enterica Typhimurium and mAdV2 with Yersinia enterocolitica, providing insights into the mechanisms that may be regulating these interactions.

Concerns:

1.The new title “Pairwise screening of viral-bacterial co-infections in vitro reveals molecular interaction points that affect pathogen proliferation and host cell viability” changed a lot compared to the last version title “Dissecting the impact of enteric viral-bacterial co-infection on the host innate immune response and its implications for pathogenicity” , the revised title is broad, I may have concern it is not suitable for the results and conclusion in the revised manuscript, and the authors should specify seven viral (murine Adenovirus (mAdV) 1, 2 and 3, and murine Norovirus (MNV) 1, 2, 3 and CR3) and six bacterial (Salmonella enterica Typhimurium, Shigella flexneri, Yersinia enterocolitica, Citrobacter rodentium, Vibrio cholerae and pathogenic Escherichia coli) enteric pathogens and two cells(RAW264.7 macrophages and iBMDMs immortalized Bone Marrow Derived Macrophages) in the manuscript with the focus (molecular interaction points).

Since the title is limited to 15 words, we have now changed it to: *‘Viral-bacterial co-infections screen in vitro reveals molecular processes affecting pathogen proliferation and host cell viability’*.

Given this restriction in length we could not accommodate the number of pathogens or their exact names in the title.

2. Considering the complexity of the intestinal environment with various types of

intestinal cells, microbiota, and intestinal contents, based on the results described *in vitro*, it should be better to discuss, to consider, test and analyse using animal models or intestinal organoids to provide *in vivo* evidences for the two newly discovered mechanisms in coinfection scenarios in the host intestine.

We agree that the intestine is an extremely complex tissue that include many different factors (cell types, microbiota, etc..) that can all affect the infection with a single pathogen or co-infections with several pathogens. Given such a complexity it is challenging to identify molecular interaction points that affect the outcome of co-infections *in vivo*. This complexity was indeed the reason why we performed the screen in a simpler model.

Analyzing the new mechanisms *in vivo* would be desirable indeed (organoid models would be not suitable as these do only include epithelial cells and not macrophages), but would go beyond the scope of the study as we do not have the license to perform co-infection experiment *in vivo*.

Reviewer #4 (Remarks to the Author):

The authors have thoughtfully addressed the points I raised; however, I do not agree that an FDR-corrected p-value for the SILAC data can be justifiably excluded from the analysis and left for other researchers to explore. Given the nature of the work and the importance of statistical robustness for accurate and reliable interpretation of the proteomics datasets, an FDR (e.g., 5%) should be set.

We thank the reviewer for the constructive feedback in the previous round and are very happy to hear that we were able to address the points that were raised. Regarding the significance values of the SILAC data, we have included the following two changes: 1) In Tables S2 and S3 (i.e. the results of our SILAC experiments, we have included a column with the FDR-corrected p-values (Benjamini-Hochberg method, $\alpha = 0.05$). 2) Throughout the manuscript, including the figure legends, we have clearly stated whether corrected or uncorrected values are displayed. We hope that this addresses the last remaining concern around data completeness for accurate interpretation.